# In Vivo Bio-Activation of JWH-175 to JWH-018: Pharmacodynamic and Pharmacokinetic Studies in Mice

**DOI:** 10.3390/ijms23148030

**Published:** 2022-07-21

**Authors:** Micaela Tirri, Raffaella Arfè, Sabrine Bilel, Giorgia Corli, Beatrice Marchetti, Anna Fantinati, Fabrizio Vincenzi, Fabio De-Giorgio, Cristian Camuto, Monica Mazzarino, Mario Barbieri, Rosa Maria Gaudio, Katia Varani, Pier Andrea Borea, Francesco Botrè, Matteo Marti

**Affiliations:** 1Section of Legal Medicine and LTTA Center, Department of Translational Medicine, University of Ferrara, 44121 Ferrara, Italy; micaela.tirri@unife.it (M.T.); raffaella.arfe@unife.it (R.A.); sabrine.bilel@unife.it (S.B.); giorgia.corli@unife.it (G.C.); beatrice.marchetti@unife.it (B.M.); fabrizio.vincenzi@unife.it (F.V.); rosamaria.gaudio@unife.it (R.M.G.); katia.varani@unife.it (K.V.); bpa@unife.it (P.A.B.); 2Department of Chemistry and Pharmaceutical Sciences, University of Ferrara, 44121 Ferrara, Italy; anna.fantinati@unife.it; 3Section of Legal Medicine, Department of Health Care Surveillance and Bioetics, Università Cattolica del Sacro Cuore, 00168 Rome, Italy; fabio.degiorgio@unicatt.it; 4A. Gemelli University Polyclinic Foundation IRCCS, 00168 Rome, Italy; 5Laboratorio Antidoping FMSI, Largo Giulio Onesti 1, 00197 Rome, Italy; loggerseason@gmail.com (C.C.); monica.mazzarino@gmail.com (M.M.); francesco.botre@unil.ch (F.B.); 6Department of Neuroscience and Rehabilitation, University of Ferrara, 44121 Ferrara, Italy; mario.barbieri@unife.it; 7University Center of Gender Medicine, University of Ferrara, 44121 Ferrara, Italy; 8Institute of Sport Science, University of Lausanne (ISSUL), Synathlon, CH-1015 Lausanne, Switzerland; 9Collaborative Center for the Italian National Early Warning System, Department of Anti-Drug Policies, Presidency of the Council of Ministers, 00186 Rome, Italy

**Keywords:** JWH-175, JWH-018, synthetic cannabinoids, toxicology, pharmacokinetics, drug metabolism, CB_1_ cannabinoid receptor

## Abstract

3-(1-Naphthalenylmethyl)-1-pentyl-1H-indole (JWH-175) is a synthetic cannabinoid illegally marketed for its psychoactive cannabis-like effects. This study aimed to investigate and compare in vitro and in vivo pharmacodynamic activity of JWH-175 with that of 1-naphthalenyl (1-pentyl-1H-indol-3-yl)-methanone (JWH-018), as well as evaluate the in vitro (human liver microsomes) and in vivo (urine and plasma of CD-1 male mice) metabolic profile of JWH-175. In vitro binding studies showed that JWH-175 is a cannabinoid receptor agonist less potent than JWH-018 on mouse and human CB1 and CB2 receptors. In agreement with in vitro data, JWH-175 reduced the fESPS in brain hippocampal slices of mice less effectively than JWH-018. Similarly, in vivo behavioral studies showed that JWH-175 impaired sensorimotor responses, reduced breath rate and motor activity, and increased pain threshold to mechanical stimuli less potently than JWH-018. Metabolic studies demonstrated that JWH-175 is rapidly bioactivated to JWH-018 in mice blood, suggesting that in vivo effects of JWH-175 are also due to JWH-018 formation. The pharmaco-toxicological profile of JWH-175 was characterized for the first time, proving its in vivo bio-activation to the more potent agonist JWH-018. Thus, it highlighted the great importance of investigating the in vivo metabolism of synthetic cannabinoids for both clinical toxicology and forensic purposes.

## 1. Introduction

Synthetic cannabinoids (SCBs) constitute the second largest group of substances reported to the United Nations Observatory on Drugs and Crime (UNODC) Early Warning Advisory (EWA) on novel psychoactive substances (NPSs) worldwide, and their use has been associated with severe negative health consequences [1]. Synthetic cannabinoids are sold as “legal” replacements for cannabis (or already controlled synthetic cannabinoids) since, at very low doses, they have similar effects to the main psychoactive substance in cannabis, Δ^9^-tetrahydrocannabinol (Δ^9^-THC), but carry additional life-threatening toxicity and can pose a high risk of severe poisoning [2,3]. SCBs have been first identified as recreational drugs of abuse in 2008 in Europe and Japan [4,5] and up to now have been detected in smoking mixtures, powders, liquids (including e-liquids), and paper impregnated with the substance (including blotters) [6].

SCBs cause more potent effects than Δ^9^-THC and produce “atypical” side effects not reported for classical Δ^9^-THC intoxication. Health-related problems associated with the use of SCBs can include severe cardiovascular toxicity (including myocardial infarction, ischemic stroke, emboli, and sudden death), acute kidney injury (AKI), severe central nervous system depression (such as rapid loss of consciousness/coma), respiratory depression, generalized tonic–clonic seizures and convulsions, hyperemesis, delirium, agitation, psychotic episodes (including first episode psychosis, paranoia, self-harm/suicide ideation), and aggressive and violent behavior [7,8,9,10,11,12]. Furthermore, preclinical studies have shown that SCBs may be genotoxic [13] and neurotoxic [14] probably by impairing mitochondrial function [15]. SCBs’ greater toxicity could be attributed to their combined pharmacodynamics (i.e., potency and biased agonism at cannabinoid receptors) [16] and pharmacokinetics of the various SCBs or their metabolites that are formed [17]. In particular, several studies revealed that SCBs are in vivo metabolized to molecules that could retain activity as agonist, antagonist, or partial agonist at CB1 and/or CB2 receptors as shown for 1-naphthalenyl(1-pentyl-1H-indol-3-yl)-methanone (JWH-018), 1-naphthalenyl(1-pentyl-1H-indol-3-yl)-methanone (JWH-073), N-((3s,5s,7s)-adamantan-1-yl)-1-(5-fluoropentyl)-1H-indazole-3-carboxamide (5F-AKB48), and (S)-N-(1-amino-3-methyl-1-oxobutan-2-yl)-1-pentyl-1H-indazole-3-carboxamide (AB-PINACA) [18,19,20,21]. However, a non-receptor-mediated mechanism has been also proposed for the toxicity of the JWH-018 main metabolite when compared to the parent drug, and for [2,3-dihydro-5-methyl-3-(4-morpholinylmethyl)pyrrolo[1,2,3-de]-1,4-benzoxazin-6-yl]-1-naphthalenyl-methanone, methanesulfonate (WIN55,212-2) in spatial memory tasks, which causes a CB-receptor-independent decrease in cholinergic activation [22,23]. SCBs and possibly their metabolites may also interact with the metabolic pathways by inhibiting cytochrome P450 (CYP450) activity [24,25,26]. Therefore, investigation of the pharmacokinetics of new SCBs and their potential metabolic activation is of great pharmaco-toxicological and forensic relevance.

The 3-(1-naphthalenylmethyl)-1-pentyl-1H-indole (JWH-175) is a naphthylmethylindole that is structurally related to JWH-018 (Figure 1), which has a methylene group that links the indole structure to the naphthyl group, instead of the carbonyl group as reported in the chemical structure of JWH-018 [27,28].

In vitro pharmacodynamics studies showed that JWH-175 retains a lower affinity for the rat CB1 receptor (Ki = 22 + 2 nM) when compared to the well-known JWH-018 (Ki = 9.5 + 4.5 nM) [28]. Despite the lack of information concerning pharmaco-toxicological profile of JWH-175, preclinical studies have shown its abuse potential on rats and inhibitory effect on locomotor activity in mice [29], and more recently, JWH-175 has been detected in oral fluid samples, suggesting its use among consumers [30].

In vitro metabolic studies, reported by previous investigators, have revealed that JWH-175 is bio-transformed to JWH-018 [31]. This bio-activation to the more potent compound could lead to important in vivo pharmaco-toxicological activity and adverse effects. Therefore, the present study is aimed to investigate if JWH-175 could be bio-transformed in vivo to JWH-018 and to characterize the pharmaco-toxicological profile of JWH-175 in comparison to its more potent analog JWH-018 in mice. In an attempt to fill a gap in the pharmaco-toxicological information about JWH-175 and based on consumer experiences about JWH-018 abuse [32], we investigated the acute systemic administration of threshold, strong, and heavy range dosages: 0.01–30 mg/kg. In vivo studies on the acute effect of JWH-175 (0.01–30 mg/kg i.p.) on sensorimotor (visual and acoustic) responses, body temperature, respiratory rate, mechanical analgesia, and motor activity (drag test) were undertaken in CD-1 mice. In order to characterize pharmaco-toxicological profile of JWH-175, its behavioral effects were monitored for over 5 h and compared with those induced by JWH-018. Furthermore, the metabolic profile of JWH-175 was investigated by liquid chromatography–triple quadrupole mass spectrometry (LC-QqQ) both in an in vitro assay (human liver microsomes) and in in vivo biological samples (urine and plasma) taken from treated animals. In vitro competition binding experiments on CD-1 murine and human CB1/CB2 receptors and electrophysiological recording of field excitatory postsynaptic potential (fEPSP) from hippocampal slices have been performed for a more complete pharmacodynamics characterization of JWH-175.

## 2. Results

### 2.1. Affinity and Potency of JWH-175 for CB1 and CB2 Receptors in Comparison to JWH-018

Competition binding experiments performed in CHO cell membranes transfected with human CB1 receptors revealed a Ki value for JWH-175 of 25.8 ± 1.9 nM, while the reference compound JWH-018 displayed a 2.71-fold higher affinity (Table 1).

JWH-175 showed a greater affinity for human CB1 receptors than for human CB2 receptors with a selectivity index (ratio between the Ki value to human CB2 and the Ki value to human CB1) of 14. As expected, JWH-018 revealed similar Ki values for the two CB receptors with a selectivity index of 0.9. Similar results were obtained when evaluating affinity values of JWH-175 and JWH-018 in mouse tissues, suggesting no species selectivity between murine and human CB receptors. Interestingly, at mouse CB1 receptors, JWH-175 showed a 5.77-fold lower affinity than JWH-018 (Table 1; Figure 2).

### 2.2. Effects of JWH-175 on Synaptic Transmission in CA1 Hippocampal Area

Vehicle did not affect excitatory transmission (fEPSP) in hippocampal slices when compared to control-untreated slices (data not shown). Application of JWH-175 on hippocampal slices induced a weak depressive effect on fEPSP (Figure 3).

Reduction in excitatory transmission reached the steady state within 60 min after drug perfusion. JWH-175 reduced fEPSP only at 1 µM (−23.7 ± 5.7; Figure 3, F(4, 27) = 2.922, *p* = 0.0395). JWH-175 failed to affect fEPSP at lower concentrations (0.01−0.3 µM). Otherwise, JWH-018 reduced fEPSP at lower concentrations compared to JWH-175, being effective at 0.1 µM (−34.7 ± 7.9; Figure 3) and at 0.3 µM (−40.4 ± 8.4 vs. vehicle; Figure 3, F(4, 27) = 8.777, *p* = 0.0001). At 1 µM, JWH-018 (−48.8 ± 7.6, Figure 3, F(4, 27) = 8.777, *p* = 0.0001) was more effective than JWH-175. Perfusion with the CB1 receptor antagonist 1-(2,4-dichlorophenyl)-5-(4-iodophenyl)-4-methyl-N-1-piperidinyl-1H-pyrazole-3-carboxamide (AM-251; 2 µM) completely prevented the inhibitory effect of JWH-175 and JWH-018 both at 1 µM (data not shown).

### 2.3. In Vivo Behavioral Studies

#### 2.3.1. Evaluation of the Visual Object Response

Visual object response did not change in vehicle-treated mice (Figure 4A), and the effect was similar to that observed in naïve untreated animals (data not shown).

Systemic administration of JWH-175 (0.01–30 mg/kg i.p.) significantly (*p* < 0.0001) and dose-dependently reduced the visual object response in mice (Figure 4A). JWH-175 produced a progressive impairment, especially at the highest doses (15–30 mg/kg i.p.), that persisted up to 5 h (Figure 4A; significant effect of treatment (F(7, 448) = 120.0, *p* < 0.0001), time (F(7, 448) = 22.42, *p* < 0.0001) and time × treatment interaction (F(49, 448) = 3.788, *p* < 0.0001)). Pretreatment with AM-251 (6 mg/kg i.p.), which alone did not alter the response in mice (Figure 4B), prevented the inhibition of visual object response induced by JWH-175 at 6 mg/kg (F(3, 28) = 16.05, *p* < 0.0001) but not that caused by 15 mg/kg dose (F(3, 28) = 33.01, *p* < 0.0001). JWH-175 appeared to be less potent in reducing visual object response, when compared to JWH-018 (Table 2; curves comparison (F(1, 92) = 892.5, *p* < 0.0001)).

#### 2.3.2. Evaluation of the Visual Placing Response

Visual placing response did not change in vehicle-treated mice (Figure 4C), and the effect was similar to that observed in naïve untreated animals (data not shown). Systemic administration of JWH-175 (0.01–30 mg/kg i.p.) dose-dependently reduced the visual placing response in mice (Figure 4A). The inhibitory effect was significant also at the lower dose tested of 0.01 mg/kg. JWH-175 induced a prolonged inhibitory effect that persisted up to 5 h (Figure 4C; significant effect of treatment (F(7, 448) = 207.5, *p* < 0.0001), time (F(7, 448) = 90.02, *p* < 0.0001) and time × treatment interaction (F(49, 448) = 6.611, *p* < 0.0001)). Pretreatment with AM-251 (6 mg/kg i.p.), which alone did not alter the response in mice (Figure 4B), prevented the inhibition of visual placing response induced by JWH-175 at 6 mg/kg but not that caused at 15 mg/kg dose (Figure 4D; significant effect of treatment (F(4, 19) = 59.65, *p* < 0.0001)). JWH-175 appeared to be less potent in reducing visual placing response when compared to JWH-018 (Table 2; curves comparison (F(1, 92) = 40.21, *p* < 0.0001)).

#### 2.3.3. Evaluation of the Acoustic Response

Acoustic response did not change in vehicle-treated mice (Figure 4E), and the effect was similar to that observed in naïve untreated animals (data not shown). Systemic administration of JWH-175 (0.01–30 mg/kg i.p.) dose-dependently reduced the visual placing response in mice (Figure 4E). The inhibitory effect of JWH-175 at 6 and 15 mg/kg was significant only after 180 min, while at the highest dose (30 mg/kg), the inhibition was significant after 60 min from drug administration and the effect persisted up to 5 h (Figure 4E; significant effect of treatment (F(7, 448) = 25.56, *p* < 0.0001), time (F(7, 448) = 17.29, *p* < 0.0001) and time × treatment interaction (F(49, 448) = 1.971, *p* = 0.0002)). Pretreatment with AM-251 (6 mg/kg i.p.), which alone did not alter the response in mice (Figure 4F), prevented the inhibition of visual placing response induced by JWH-175 at 6 mg/kg (F(3, 28) = 28.84, *p* < 0.0001) and at 15 mg/kg (F(3, 28) = 9.746, *p* = 0.0001). JWH-175 appeared to be less potent in reducing acoustic response when compared to JWH-018 (Table 2; curves comparison (F(1, 92) = 212.5, *p* < 0.0001)).

#### 2.3.4. Evaluation of Breath Rate

Breath rate did not change in vehicle-treated mice over the 5 h observation (Figure 5A), and the effect was similar to that observed in naïve untreated animals (data not shown).

Systemic administration of JWH-175 (0.01–30 mg/kg i.p.) dose-dependently reduced the breath rate in mice (Figure 5A; significant effect of treatment (F(7, 448) = 73.23, *p* < 0.0001), time (F(7, 448) = 28.72, *p* < 0.0001) and time × treatment interaction (F(49, 448) = 1.850, *p* = 0.0007)). The inhibitory effect was significant at 0.1 mg/kg after 120 min and the effect persisted up to 5 h. Pretreatment with AM-251 (6 mg/kg i.p.), which alone did not alter the respiratory rate in mice (Figure 5B), prevented the reduction in breath rate induced by JWH-175 at 6 mg/kg (F(3, 28) = 10.51, *p* = 0.0001) and at 15 mg/kg (F(3, 28) = 30.46, *p* < 0.0001). ED50 value of JWH-175 was 1.7 + 0.19 mg/kg (Table 2).

#### 2.3.5. Evaluation of Core Temperature

Core temperature did not change in vehicle-treated mice over the 5 h observation (Figure 5C), and the effect was similar to that observed in naïve untreated animals (data not shown). Core temperature is reduced by systemic administration of JWH-175 (0.01–30 mg/kg; i.p.) (Figure 5C; significant effect of treatment (F(7, 392) = 38.60, *p* < 0.0001), time (F(6, 392) = 9.260, *p* < 0.0001) but not time × treatment interaction (F(42, 392) = 0.7318, *p* = 0.8925)). The inhibitory effect was mild and transient at 6 mg/kg at 200 min, while prolonged at higher doses (15–30 mg/kg), and the effect persisted up to 5 h. Pretreatment with AM-251 (6 mg/kg; i.p.), which alone did not alter the core temperature in mice (Figure 5D), prevented the core temperature decrease induced by JWH-175 at 6 mg/kg (F(3, 23) = 16.41, *p* < 0.0001) and at 15 mg/kg (F(3, 23) = 140.6, *p* < 0.0001). JWH-175 appeared to be less potent in reducing core temperature of mice, when compared to JWH-018 (Table 2; curves comparison (F(1, 92) = 63.63, *p* < 0.0001)).

#### 2.3.6. Evaluation of Pain Induced by a Mechanical Stimulus

Tail pinch test response did not change in vehicle-treated mice over the 5 h observation (Figure 6A), and the effect was similar to that observed in naïve untreated animals (data not shown).

Systemic administration of JWH-175 (0.01–30 mg/kg; i.p.) dose-dependently increased the threshold to acute mechanical pain stimulus in mice (Figure 6A; significant effect of treatment (F(7, 392) = 108.6, *p* < 0.0001), time (F(6, 392) = 38.69, *p* < 0.0001) and time × treatment interaction (F(42, 392) = 2.668, *p* < 0.0001)). The analgesic effect increases over time and at the maximum dose of 30 mg/kg reaches the maximum effect at 265 min. Pretreatment with AM-251 (6 mg/kg i.p.), which alone did not alter the threshold to acute mechanical pain response in mice (Figure 6B), prevented the inhibitory effect induced by JWH-175 at 6 mg/kg (F(3, 28) = 27.93, *p* < 0.0001) but not that induced at 15 mg/kg (F(3, 28) = 43.76, *p* < 0.0001). JWH-175 appeared to be statistically less potent than JWH-018 in increasing the pain threshold to acute mechanical stimuli (Table 2; curves comparison (F(1, 92) = 37.476, *p* = 0.0455)).

#### 2.3.7. Evaluation of Number of Steps

The number of steps remained unchanged in vehicle-treated mice over the 5 h observation (Figure 6C), and the effect was similar to that observed in naïve untreated animals (data not shown). Systemic administration of JWH-175 (0.01–30 mg/kg; i.p.) facilitated at low (0.1 mg/kg) and inhibited at higher doses (6–30 mg/kg) the motor activity of mice in the drag test (Figure 6C; significant effect of treatment (F(7, 448) = 66.33, *p* < 0.0001), time (F(7, 448) = 9.592, *p* < 0.0001) and time × treatment interaction (F(49, 448) = 2.806, *p* < 0.0001)). Specifically, at 0.1 mg/kg JWH-175 induced a prolonged facilitation that lasted up to 280 min, while at 6 mg/kg the inhibitory effect was transient and significant only at 160 min. At 15 mg/kg the inhibition was significant at 160 min and was maintained up to 340 min, while at the highest dose (30 mg/kg) the inhibition of motor activity appeared as early as 75 min and was maintained up to 340 min. Pretreatment with AM-251 (6 mg/kg; i.p.), which alone did not alter the motor response in mice (Figure 6D), prevented the inhibitory effect induced by JWH-175 at 6 mg/kg (F(3, 28) = 3.215, *p* = 0.0379) but not that induced at 15 mg/kg (F(3, 28) = 8.313, *p* = 0.0004). Unlike JWH-018 (ED50 = 2.4 + 0.12 mg/kg), which reduces number of steps at all doses, ED50 calculated on inhibitory effect induced by the highest doses of JWH-175 tested was 10.9 + 0.12 mg/kg (Table 2). JWH-175 appeared to be statistically less potent than JWH-018 in reducing the stepping activity in the drag test (Table 2; curves comparison F(1, 76) = 11.71, *p* = 0.0010).

#### 2.3.8. Results of Metabolic and Behavioral Studies

##### Plasmatic Profile and Correlations with Behavioral Responses

Analysis of the plasmatic samples taken from mice treated with JWH-175 (10 mg/kg i.p.) revealed the presence of JWH-018 as the main metabolite of JWH-175. These results unequivocally confirm the in vivo formation of JHW-018. The plasmatic concentration of JWH-018 showed a maximum at 180 min after administration of the parent compound, while JWH-175 itself was detected only in the first 30 min from the administration (Figure 7).

Analysis of the plasma samples taken from mice treated with JWH-018 (10 mg/kg; i.p.) revealed that under our experimental conditions the peak plasmatic concentration of JWH-018 showed a maximum at 30 min, while after a low decrease following the peak concentration, plasmatic concentrations of JWH-018 remained quite stable for the following 180–300 min. Nevertheless, JWH-018 plasmatic levels remain significantly higher than those observed after systemic administration of JWH-175 at 10 mg/kg (Figure 8A; significant effect of treatment (F(2, 60) = 154.2, *p* < 0.0001), time (F(3, 60) = 56.69, *p* < 0.0001) and time × treatment interaction (F(6, 60) = 21.70, *p* < 0.0001)). Results are expressed as the average ng/mL calculated from the concentration extrapolated from the calibration curve (R > 0.990) for each mouse collecting point (i.e., 30, 180, and 300 min).

Plasma concentrations of JWH-018, derived from JWH-175 or JWH-018 administration, are directly correlated with behavioral and physiological changes in mice (Figure 8).

In particular, visual object response (Figure 8B; Pearson’s r = −0.9815, *p* < 0.0001), visual placing response (Figure 8C; Pearson’s r = −0.9140, *p* < 0.0001), acoustic response (Figure 8D; Pearson’s r = −0.7694, *p* < 0.0001), body temperature (Figure 8E; Pearson’s r = −0.8647, *p* < 0.0001), breath rate (Figure 8F; Pearson’s r = −0.9126, *p* < 0.0001), mechanical analgesia (Figure 8G; Pearson’s r = 0.8431, *p* < 0.0001), and motor activity (Figure 8H; Pearson’s r = −0.9193, *p* < 0.0001) were significantly correlated to JWH-018 (from injection of JWH-175) plasma concentrations. Similarly, visual object response (Figure 8B; Pearson’s r = −0.9670, *p* < 0.0001), visual placing response (Figure 8C; Pearson’s r = −0.9682, *p* < 0.0001), acoustic response (Figure 8D; Pearson’s r = −0.9379, *p* < 0.0001), breath rate (Figure 8F; Pearson’s r = −0.9618, *p* < 0.0001), mechanical analgesia (Figure 8G; Pearson’s r = 0.9873, *p* < 0.0001), and motor activity (Figure 8H; Pearson’s r = −0.9607, *p* < 0.0001) were significantly correlated to JWH-018 (from injection of JWH-018 itself) plasma concentrations. Otherwise, body temperature (Figure 8E; Pearson’s r = −0.6265, *p* = 0.0011) was not significantly correlated to JWH-018 (from injection of JWH-018 itself) plasma concentrations.

##### Urinary Excretion

Samples from the mice excretion study were analyzed, and the results obtained showed the formation of mono-, di-hydroxylated, and dehydrogenated-mono-OH metabolites of JWH-018 (Figure 9A).

To confirm the previously reported observation, the phase I metabolites of JWH-018 were enzymatically synthesized by incubating a standard solution of JWH-018 in the presence of human liver microsomes (HLMs) (Figure 9D–F). The in vitro metabolic profile was compared with those obtained from the analysis of the urine samples collected from mice after JWH-175 administration. The results showed the formation of ten metabolites of JWH-018 in common between the in vitro and in vivo samples: three di-hydroxylated metabolites (M1–M3, formed by CYP3A4 and CYP3A5 isoenzymes), three mono-hydroxylated metabolites (M8-M10, formed mainly by CYP3A4 and CYP3A5 and to a lesser degree by CYP2C9, CYP2C19, and CYP2D6 isoenzymes), and four dehydrogenated-mono-OH metabolites (M4–M7, formed by CYP3A4 and CYP3A5 isoenzymes) were identified in both studies (Figure 9). These observations confirm the results obtained in plasma (see again Figure 7): after administration, JWH-175 is extensively bio-transformed to JWH-018 by CYP3A4 and CYP3A5 isoenzymes.

## 3. Discussion

The present study for the first time compared the in vitro and in vivo pharmacodynamic and pharmacokinetic activity of the synthetic cannabinoid JWH-175 with those of its analog JWH-018. In vitro binding studies show that JWH-175 retains nanomolar affinity for both CD-1 murine and human CB1 and CB2 receptors, albeit less marked when compared to JWH-018 in particular on CB2 receptors. In vitro electrophysiological recording shows that JWH-175 is less potent and effective than JWH-018 in reducing fEPSP in hippocampal slices. Similarly, in in vivo studies, JWH-175 was less potent and effective than JWH-018 in impairing visual and acoustic sensorimotor responses, inducing hypothermia, reducing breath rate, altering motor performance, and promoting mechanical analgesia in mice. A relevant aspect of this study is that we firstly demonstrated that JWH-175 is rapidly metabolized in vivo to the more potent synthetic cannabinoid JWH-018 and that its bio-transformation into JWH-018 is directly correlated with the behavioral and physiological changes occurring in mice.

From a structural point of view, JWH-175 is a naphthylmethylindole [27,28] closely related to JWH-018 but where the ketone bridge is replaced by a simpler methylene group (Figure 1). Despite the close similarity in structure between the two synthetic cannabinoids, JWH-175 showed less affinity and potency than JWH-018 for either human or murine CB1 receptors. Furthermore, a preference of JWH-175 for the CB1 receptor has been observed, while JWH-018 had a similar affinity for both CB1 and CB2 receptors (Table 1). Indeed, the different affinity of JWH-175 compared to JWH-018 was even more evident towards CB2 than CB1 receptors both in human and murine preparations. This pharmacodynamic profile is reflected in the lower potency and efficacy of JWH-175 compared to JWH-018 in in vitro electrophysiology studies. In fact, JWH-175 is about 10 times less potent than JWH-018 in inhibiting fEPSP from CA1 hippocampal slices, showing less efficacy in the dose range studied (0.01–1 µM). The present in vitro results demonstrate that JWH-175 depressed the synaptic excitatory transmission (fEPSP) in mouse hippocampal slices by CB1 receptor activation (blockade by AM251), similarly to what was previously reported for the JWH-018 [35,36] and its halogenated derivatives [35]. This electrophysiological effect is in line with findings demonstrating that CB1 receptor agonists induce, in the hippocampus, depressive effects on in vitro synaptic glutamatergic transmission [37,38,39,40,41,42,43,44], thus possibly impairing in vitro mechanisms of synaptic long-term potentiation and in vivo memory acquisition [38].

### 3.1. Behavioral Studies

In vivo experiments showed that JWH-175 impairs visual and acoustic sensorimotor responses, causes hypothermia, reduces breath rate, alters motor performance, and promotes mechanical analgesia in mice as reported for other JWH-type SCBs [33,34,45,46,47,48,49]. However, behavioral (motor impairment), physiological (hypothermia, analgesia, breath rate), and sensorimotor (visual, acoustic) responses induced by JWH-175 appear to be less potent (Table 2) than those induced by JWH-018 [33,34]. Thereby, these pieces of evidence suggest that the in vivo efficacy and potency of these compounds can possibly depend on both pharmacodynamic (i.e., receptor affinity) and pharmacokinetic (i.e., metabolism) benchmarks [50].

JWH-175 induced a deep sensorimotor impairment affecting the visual and acoustic responses of mice. As previously assumed for other SCBs [15,33,34,51], JWH-175 could affect the visual information processing of rodents by acting on CB1 receptors expressed in the thalamocortical–striatal visual circuitry [52,53,54]. This assumption is supported by further studies showing that visual information in mice is elaborated in a subpopulation of neurons selectively localized in the dorsomedial striatum [55]. However, CB1 receptors are also localized in the retina, cornea, iris, and choroid, thus possibly contributing to the effect induced by SCBs on ocular functions and vision itself [9,56].

Furthermore, it has been demonstrated that acoustic startle reflex is induced by the activation of three serially connected structures that involve the activation of the dorsal cochlear nucleus [57]. Thereby, SCBs likely act on the CB1 receptors in the presynaptic terminals of parallel fibers in the dorsal cochlear nucleus and alter the acoustic response of mice [58]. In line with this evidence, our previous work showed that JWH-018 halogenated derivatives impaired the prepulse inhibition (PPI) responses in mice, confirming the detrimental effect of such compounds on sensory gating functions [46].

Systemic administration of JWH-175 moreover induced a deep and long-lasting decrease in the breath rate of mice, especially at the highest dose tested (30 mg/kg). This agrees with previous studies, showing that Δ9-THC [59,60,61] and SCBs induced acute respiratory depression in rodents [51,62]. Such a decrease in respiratory rate could be related to the direct action of SCBs on the central nervous system (CNS), in which CB1 receptors are highly expressed [63]. However, it has been demonstrated that CB1 receptors found on airway nerves mediated effects on bronchial responsiveness of rodents [64], suggesting that an additional peripheral action should not be ruled out.

Similarly, JWH-175 lowered the core temperature of mice, inducing prolonged hypothermia at the highest dose administered (30 mg/kg). Previous studies have already shown the hypothermic effect of JWH-018 [33,65,66,67] and its halogenated derivatives [44], in line with our results. As formerly pointed out, SCBs such as JWH-175 and JWH-018 likely induce a CB1 receptor-mediated effect on body temperature by acting in the preoptic anterior hypothalamic area of the CNS [67,68].

Increased pain threshold to acute mechanical stimulus has been observed in mice starting from the lowest doses tested (0.1 mg/kg), as previously shown on JWH-type compounds [33] and other SCBs [15,51,69,70]. It has been pointed out that WIN 55,212-2 possibly induced analgesia via acting on cannabinoid receptors and thus affecting stimulus-evoked activity in the ventral posterolateral nucleus of the thalamus [70]. Particularly, more recent studies have highlighted that CB1 receptors are located in mid- and hindbrain regions, and the spinal cord of rodents [52,71,72] and may have an important role in altering the pain threshold. In fact, the contribution of both central [73] and peripheral CB1 receptors to the antinociceptive effect has been demonstrated [74,75,76].

Noteworthily, the administration of JWH-175 at low (0.1 mg/kg) and higher doses (6–30 mg/kg) facilitated and inhibited, respectively, the stepping activity of mice in the drag test. A biphasic profile on motor performance has been previously reported in the modulation of spontaneous locomotor activity in rodents by anandamide [77], Δ9-THC [34], WIN 55,212-2 [78], JWH-018, JWH-073 and 1-(1-pentyl-1H-indol-3-yl)-2-(2-methoxyphenyl)-ethanone (JWH-250) [34,45], 1-pentyl-*N*-tricyclo [3.3.1.13,7]dec-1-yl-1*H*-indazole-3-carboxamide (AKB48) [15], and *N*-adamantyl-1-fluoropentylindole-3-carboxamide (STS-135) [79], suggesting that this modulation is typical of the cannabinoid system and not of a single molecule class [80]. In particular, it has been shown that the increase in spontaneous locomotor activity induced by JWH-018 can be related to a CB1 receptor and dopamine (DA)-mediated mechanism [81]. Furthermore, previous studies showed that Δ9-THC and SCBs may regulate motor activity by acting on CB1 receptors located in the cerebellum and basal ganglia, thus affecting dopaminergic motor circuits or central glutamate neurotransmission [82,83].

As previously pointed out for JWH-018 and other JWH-type SCBs [33,34,45,46], our results strengthen the hypothesis that effects induced by JWH-175 up to 6 mg/kg were fully dependent on CB1 receptor stimulation. Indeed, they were completely prevented by the administration of the selective CB1 receptor antagonist/inverse agonist AM-251. The latter also prevented acoustic impairment, respiratory depression, and hypothermia induced by a higher dose of JWH-175 (15 mg/kg), confirming that such effects were CB1 receptor-mediated. However, pretreatment with AM-251 did not abolish the impairment of the visual sensorimotor responses and stepping activity (drag test) or the increased mechanical analgesia induced by the same dose of JWH-175. Relying on this study, we cannot exclude that a higher dose of AM-251 would have been required to observe the full prevention of the effect induced by 15 mg/kg of JWH-175.

Nevertheless, this different response profile could be due to the possible interaction between SCBs and non-cannabinoid receptors. Indeed, previous studies revealed that SCs can exert their action via possibly acting on many different targets such as calcium, potassium, and sodium channels [84,85]; G protein-coupled receptor 55 (GPR55) [86]; peroxisome proliferator-activated receptors (PPARs) [84]; and transient receptor potential channels type vanilloid (TRPV1) and ankyrin (TRPA1) [87]. Furthermore, it has been suggested that small structural disparities between ligands (carbonyl vs. carboxyl bridge) may induce different active receptor conformations to differentially activate downstream signaling pathways (termed “biased agonism” or “functional selectivity”), which may have implications for pharmaco-toxicity in vivo [16]. Therefore, further studies should be carried out to better understand which pharmacological mechanism could be related to certain lingering effects induced by the highest doses tested of JWH-175.

### 3.2. JWH-175 and JWH-018 Profile in Plasma

The analytical method used to determine the compounds under investigation in biological matrices was validated in terms of specificity, sensitivity, limits of detection and quantification, linearity, recovery, matrix effect, accuracy, and precision. No significant interferences were observed at the retention times of the compounds under investigation and ISTD in the chromatograms of the corresponding ion transitions in negative samples. Limits of detection and quantification were 0.5 and 5 ng/mL, respectively, for all the compounds considered. The analytical method developed was shown to be linear in the studied range of concentrations (5–1000 ng/mL) with r^2^ exceeding 0.95. Recoveries greater than 65% were measured, and no correlation was observed between extraction recoveries and concentrations. The matrix effect was found to be lower than 30%. Regarding intraday and intermediate precisions, RSDs were below 20% for the lowest QC samples and below 15% for the other two QC samples. As for accuracy, relative errors were better than 20% for the lowest QC samples and better than 15% for the other two QC samples. According to the results reported above, the developed method was used for an accurate quantitation of the compounds under investigation. The results obtained by analyzing plasma and urine samples collected from mice after administration of JWH-175 showed the bio-transformation of JWH-175 to the more potent cannabimimetic JWH-018, confirming the results reported by Fietzke et al. [31] after in vitro investigation. However, in contrast to the data reported by Fietzke et al., in which JWH-018 was formed in a low amount [31], the results obtained after analysis of plasma and urine samples revealed that the bio-transformation of JWH-175 to JWH-018 was rapid and extensive: (i) JWH-175 was not detectable in the plasma of mice after 30 min, (ii) JWH-018 was the most abundant compound detected in all the plasma samples analyzed, and finally (iii) ten phase I metabolites of JWH-018 (M1-M10, see again Figure 9) were detected in the urine samples collected after administration of JWH-175.

The above findings indicate that behavioral and physiological alterations observed after JWH-175 administration were prevalently dependent on JWH-018 formation in plasma. This hypothesis correlates with the increase in efficacy of JWH-175 in in vivo assays with respect to in vitro electrophysiology recording of fEPSP in hippocampal slices where the bioactivation of JWH-175 to JWH-018 is to be considered minimal. In fact, nevertheless, it has been demonstrated that cytochrome P450 enzymes are expressed in rodent brain [88,89] and in the mouse hippocampus [90] in adequate amounts to catalyze the bio-transformation of a variety of endogenous compounds, xenobiotics, and psychoactive drugs [91]; in our in vitro electrophysiological set-up, the role of cytochromes on the metabolism of JWH-175 is to be considered minimal or absent. This is due to the fact that hippocampal slices are in continuous superfusion inside the recording bath, and the possibility that JWH-175 is bioactivated to JWH-018 capable of activating the cannabinoid receptors in the tissue of the slice is remote.

## 4. Materials and Methods

### 4.1. Animals

Experimental protocols routinely adopted in our laboratory for the pharmaco-toxicological characterization of NPSs were carried out on a mouse model. This was particularly suitable both for the choice of dosages and for the execution of the provided experimental protocols [15,33]. Male ICR (CD-1) mice weighing 30–35 g (Centralized Preclinical Research Laboratory, University of Ferrara, Italy) were group-housed (5 mice per cage; floor area per animal was 80 cm^2^; minimum enclosure height was 12 cm), exposed to a 12:12-h light–dark cycle (light period from 6:30 AM to 6:30 PM) at a temperature of 20–22 °C and humidity of 45–55% and were provided ad libitum access to food (Diet 4RF25 GLP; Mucedola, Settimo Milanese, Milan, Italy) and water. The experimental protocols were in accordance with the European Communities Council Directive of September 2010 (2010/63/EU), a revision of the Directive 86/609/EEC, and were approved by the Ethics Committee of the University of Ferrara and by the Italian Ministry of Health (license No. 956/2020-PR and license No. 223/2021-PR, CBCC2.46.EXT.21). Moreover, adequate measures were taken to reduce the number of animals used and their pain and discomfort according to the ARRIVE guidelines.

### 4.2. Drug Preparation and Dose Selection

JWH-175 and JWH-018 were purchased from LGC Standards (LGC Standards, Milan, Italy), while AM-251 was from Tocris (Tocris, Bristol, United Kingdom). In in vitro electrophysiological studies, JWH-175, JWH-018, and AM-251 were dissolved in ethanol. The CB1 receptor-preferring antagonist/inverse agonist AM-251 (2 µM) was perfused 30 min before JWH-175 and JWH-018 administration. The final ethanol concentration was below 1\10,000 in any test. In in vivo studies, all compounds were initially dissolved in absolute ethanol (final concentration: 5%) and Tween 80 (2%) and brought to the final volume with saline (0.9% NaCl). The solution made with absolute ethanol, Tween 80, and saline was also used as the vehicle. AM-251 (6 mg/kg) was administered 20 min before JWH-175 injections. Drugs were administered by an intraperitoneal route at a volume of 4 uL/gr. Doses of JWH-175 were chosen based on our previous study on synthetic naphtoylindole derivatives [33,34,45,46]. Moreover, given that the main purpose of this study is to test whether JWH-175 metabolizes on JWH-018 in vivo, the choice of acute over protracted administration is justified. In fact, repeated administration of SCBs may possibly lead to altered metabolic patterns [25].

### 4.3. In Vitro Studies

#### 4.3.1. Mouse Brain and Spleen Membrane Preparation

To evaluate the affinity of JWH-175 for murine CB1 and CB2 receptors, membranes from the mouse brain and spleen were used, respectively. Following excision from mice, tissues were suspended in 50 mM Tris HCl, pH 7.4, at 4 °C. The mouse brain and spleen tissues were homogenized with a Polytron and subsequently centrifuged for 10 min at 2000× *g*. The resulting supernatants were filtered and centrifuged for 20 min at 40,000× *g*, and the pellets were used for competition binding experiments [92].

#### 4.3.2. Cell Culture and Membrane Preparation

Chinese hamster ovarian (CHO) cells transfected with human CB1 or CB2 receptors (Perkin Elmer Life and Analytical Sciences, Melville, NY, USA) were grown adherently and maintained in Ham’s F12 containing 10% fetal bovine serum, penicillin (100 U/mL), streptomycin (100 µg/mL), and Geneticin (G418, 0.4 mg/mL) at 37 °C in 5% CO_2_/95% air. To obtain membranes, cells were washed with phosphate-buffered saline (PBS) and scraped off with ice-cold hypotonic buffer (5 mM Tris HCl, 2 mM EDTA, pH 7.4). The cell suspension was homogenized with a Polytron and then centrifuged for 30 min at 40,000× *g*. The membrane pellet was suspended in 50 mM Tris HCl buffer (pH 7.4) containing 2.5 mM ethylenediaminetetraacetic acid (EDTA), 5 mM MgCl_2_, and 0.5 mg/mL bovine serum albumin (BSA) for CB1 receptors or in 50 mM Tris HCl (pH 7.4), 1 mM EDTA, 5 mM MgCl_2_, and 0.5% BSA for CB2 receptors [92].

#### 4.3.3. [3H] CP-55,940 Competition Binding Assays

Competition binding experiments were carried out by incubating 0.5 nM [3H]-CP-55,940 (Perkin Elmer Life and Analytical Sciences, Waltham, MA, USA) and different concentrations of the tested compounds for 90 or 60 min at 30 °C for CB1 or CB2 receptors, respectively. For human cannabinoid receptors, membranes obtained from CHO cells transfected with human CB1 or CB2 receptors (2 µg protein/100 µL) were used. Competition binding experiments at murine cannabinoid receptors were performed with mouse brain membranes (40 µg protein/100 µL) or with mouse spleen membranes (80 µg protein/100 µL) for CB1 receptors or CB2 receptors, respectively. Non-specific binding was determined in the presence of 1 μM WIN 55,212-2 [92]. Bound and free radioactivity were separated by filtering the assay mixture through Whatman GF/C glass fiber filters using a Brandel cell harvester (Brandel Instruments, Unterföhring, Germany). The filter-bound radioactivity was counted using a Packard Tri Carb 2810 TR scintillation counter (Perkin Elmer Life and Analytical Sciences, Waltham, MA, USA).

#### 4.3.4. Cyclic AMP Assays

CHO cells transfected with human CB1 or CB2 receptors were washed with PBS, detached with trypsin, and centrifuged for 10 min at 200× *g*. The pellet containing 1 × 106 cells/assay was suspended in 0.5 mL of 150 mM NaCl, 2.7 mM KCl, 0.37 mM NaH_2_PO_4_, 1 mM MgSO_4_, 1 mM CaCl_2_, 5 mM HEPES, 10 mM MgCl_2_, and 5 mM glucose, pH 7.4, at 37 °C. Cells were pre-incubated with 0.5 mM of the phosphodiesterase inhibitor 4-(3-butoxy-4-methoxybenzyl)-2-imidazolidinone (Ro 20-1724) for 10 min in a shaking bath at 37 °C. The potency of the examined compounds was studied in the presence of forskolin 1 µM. The reaction was terminated by the addition of cold 6% trichloroacetic acid (TCA), and the final aqueous solution was tested for cyclic AMP levels using a competition protein binding assay [92].

#### 4.3.5. Data Analysis

The protein concentration was determined according to a Bio-Rad method with bovine serum albumin as a reference standard. Inhibitory binding constants, Ki, were calculated from the IC50 values according to the Cheng and Prusoff equation:Ki = IC50/(1 + [C*]/KD*)
where [C*] is the concentration of the radioligand and KD* its dissociation constant. Functional experiments were analyzed by non-linear regression analysis using the equation for a sigmoid concentration–response curve (GraphPad Prism, San Diego, CA, USA). All the data are expressed as the mean ± SEM of three independent experiments.

### 4.4. In Vitro Electrophysiological Studies in Hippocampal Slices

#### 4.4.1. Tissue Preparation

The hippocampal transverse slice model was used to evaluate the acute effects of JWH-175 and JWH-018 on synaptic excitatory transmission. Procedures for tissue preparation are the same as those described by Ossato et al., 2016 [45]. In brief, after mouse decapitation, the head was immediately chilled in refrigerated artificial cerebrospinal fluid (aCSF, 0 °C) of the following composition (in mM): NaCl, 126; KCl, 2; KH_2_PO_4_, 1.25; NaHCO_3_, 26; MgSO_4_, 2.0; CaCl_2_, 2.5; d-glucose, 10. After 1 min, the brain was extracted to isolate the hippocampus and then placed in ice-cold artificial cerebrospinal fluid (aCSF). All solutions were saturated with 95% O_2_/5% CO_2_. Transverse hippocampal slices (425 μm thick) were cut with a tissue chopper and then placed for almost 90 min in a Haas-style incubation chamber for recovery before recording.

#### 4.4.2. Electrophysiological Recording

A single slice was transferred into a submerged-type recording chamber (3 mL total volume) and continuously superfused (3.0 mL/min) with warmed (32–33 °C) aCSF. WINLTP 2.10 computer software [93] was used for stimulus triggering, PC recording, and analysis. Synaptic responses of CA1 pyramidal neurons were elicited by electrical stimulation of the Schaffer collateral/commissural pathway. Single stimulation pulses (80 µs duration; 0.05 Hz) were delivered by mean of a concentric bipolar electrode (o.d. 125 μm, FHC, USA) connected to a Grass S11 stimulator driving a PSIU6 constant current stimulus isolation unit. To record the fEPSPs, borosilicate glass electrodes fabricated using a microelectrode puller (P97, Sutter Instruments) and filled with aCSF (1.5 ± 0.5 MΩ), inserted in the distal third of the stratum radiatum and 200–300 µm away from stimulating electrodes, were used. The maximal fEPSPs response was reached by adjusting the recording electrode depth. Recorded potentials were amplified (Axoprobe 1A DC-coupled—Cyberamp 320, Molecular Devices, Sunnyvale, CA, USA) and filtered (5.0 kHz) prior to A/D conversion. Upon 10 min of stable synaptic response, a stimulus–response curve (SRC) was generated as previously described [45] to extrapolate a stimulation intensity evoking 70% of maximal fEPSP, to be used throughout the experiment. JWH-175, JWH-018, and AM-251 (dissolved in EtOH) were added from a stock solution to the reservoir and bath applied via perfusion line. To investigate whether the vehicle had any effect on synaptic activity, the superfusion inlet was switched to a reservoir containing aCSF plus the amount of vehicle present for the corresponding drug concentration (sham application), before switching to the solution also including the drug, for comparison. All reservoirs and tubing were on glass or Teflon to avoid the capturing of cannabinoids by plastic or silicon parts.

#### 4.4.3. Data Analysis

We analyzed data from in vitro recordings as previously described [45]. In brief, the fEPSP slope was measured to calculate drug effects on electrically induced synaptic excitatory transmission. Effect of treatment was considered steady state when stable +/− 5% for 10 min.

### 4.5. In Vivo Behavioral Studies

For the overall study, 132 mice were used. In sensorimotor tests for JWH-175 experiments, for each treatment (vehicle or 0.01–30 mg/kg JWH-175 doses), 8 mice were used (total mice used: 56); for JWH-175 with AM-251 experiments, for each treatment, 8 mice were used (total mice used: 16); for JWH-018 and JWH-175 blood collection and behavioral experiments, 6 mice were used at each time point (total mice used: 48); for urinary excretion studies, for each treatment (vehicle or JWH-175 10 mg/kg), 6 mice were used (total mice used: 12).

In the present study, the effect of JWH-175 on pharmaco-toxicological responses was investigated using a battery of behavioral tests widely used in studies of “safety-pharmacology”, routinely adopted in our laboratory, for the preclinical characterization of new molecules in rodents [33,34,46,94,95,96]. The motor responses of the animal to different visual and acoustic stimuli were evaluated according to the procedure described in our previous studies [33,34,45,46]. To reduce the number of animals used, mice were evaluated in functional observational tests carried out in a consecutive manner according to the following time scheme: Observation of visual object and acoustic responses and breath rate were measured at 0, 10, 30, 60, 120, 180, 240, and 300 min after injections. Visual placing response was measured at 0, 15, 35, 70, 125, 185, 245, and 305 min after injection. Core temperature was measured at 25, 45, 85, 140, 200, 260, and 320 min after injection. The tail pinch test was measured at 35, 55, 90, 145, 205, 265, and 325 min after injection, and the valuation of number of steps in the drag test was measured at 45, 75, 105, 160, 220, 280, and 340 min after injection.

Behavioral tests were conducted in a thermostated (temperature: 20–22 °C, humidity: 45–55%) and light-controlled (150 lux) room with a background noise of 40 ± 4 dB. The apparatus for the visual object, acoustic, and tactile sensorimotor tests consisted of an experimental chamber (350 × 350 × 350 mm) with black methacrylate walls and a transparent front door. During the week before the experiment, each mouse was placed in the box and handled (once a day) every other day, i.e., 3 times, to allow it to acclimate to both the environment and the experimenter. To avoid mouse olfactory cues, cages were carefully cleaned with a dilute (5%) ethanol solution and rinsed with water. All experiments were performed between 8:30 AM and 2:00 PM and conducted blindly by trained observers working in pairs [34]. The behavior of mice was videotaped using a camera (B/W USB Camera day&night with varifocal lens; Ugo Basile, Gemonio, Italy) placed at the top or on one side of the box and analyzed off-line by a different trained operator.

#### 4.5.1. Evaluation of the Visual Responses

The visual response was verified by two behavioral tests which evaluated the ability of the animal to capture visual information when the animal is either stationary (the visual object response) or moving (the visual placing response).

The visual object response test was used to evaluate the ability of the mouse to see an object approaching from the front (frontal view) or the side (lateral view) that typically induces the animal to shift or turn the head, bring the forelimbs in the position of “defense” or retreat from it. For the frontal visual response, a white horizontal bar was moved frontally to the mouse head, and the maneuver was repeated 3 times. For the lateral visual response, a small dentist’s mirror was moved into the mouse’s field of view in a horizontal arc, until the stimulus was between the mouse’s eyes. The procedure was conducted bilaterally [34,95] and was repeated 3 times. The score assigned was 1 if there was a reflection in the mouse movement or 0 if it was not present. The total value was calculated by adding the scores obtained in the frontal with those obtained in the lateral visual object response test (overall score: 9).

The visual placing response test was performed using a modified tail suspension apparatus able to bring down the mouse towards the floor at a constant speed of 10 cm/s [34]. Briefly, CD-1 mice were suspended 20 cm above the floor by adhesive tape that was placed approximately 1 cm from the tip of the tail. The downward movement of the mouse was videotaped using a camera (B/W USB Camera day&night with varifocal lens; Ugo Basile, Italy) placed at the base of the tail suspension apparatus. Movies were analyzed off-line by a trained operator who was unaware of the drug treatments performed. The frame-by-frame analysis allows evaluating the beginning of the reaction of the mouse while it was approaching the floor. The first movement of the mouse when it perceives the floor is the extension of the front legs. When the mouse started the reaction, an electronic ruler evaluated the perpendicular distance in millimeters from the eyes of the mice to the floor. Untreated control mice typically perceive the floor and prepare for contact at a distance of about 20.4 ± 3.2 mm.

#### 4.5.2. Evaluation of the Acoustic Response

Acoustic response measures the reflex of the mouse in response to an acoustic stimulus produced behind the animal [34,46]. In particular, four acoustic stimuli of different intensities and frequencies were tested: a snap of the fingers (four snaps repeated in 1.5 s), a sharp click (produced by a metal instrument; four clicks repeated in 1.5 s), an acute sound (produced by an audiometer; frequency: 5.0–5.1 kHz), and a severe sound (produced by an audiometer; frequency: 125–150 Hz). Each test was repeated 3 times. The score assigned was 1 if there was a response or 0 if it was not present, for a total score of 3 for each sound. The acoustic total score was calculated by adding the scores obtained in the four tests (overall score: 12). The background noise (about 40 ± 4 dB) and the sound from the instruments were measured with a digital sound level meter.

#### 4.5.3. Evaluation of Breath Rate

The experimental protocol for the detection of respiratory parameters in this study provides for monitoring of the animal awake, freely moving, with non-invasive and minimal handling [95]. The animal was left free in a cage, and the respiration patterns of the mice were videotaped using a camera (B/W USB Camera day&night with varifocal lens; Ugo Basile, Italy) placed above the observation cage. A trained operator who did not know the drug treatments performed analysis of movies off-line. The frame-by-frame analysis allows a better evaluation of the breath rate of the mouse evaluated through the count of about 264 ± 13 breaths per minute (bpm).

#### 4.5.4. Evaluation of Core Temperature

To better assess the effects of the ligands on thermoregulation, we measured changes in the core (rectal) temperature. Rectal body temperature was used as an index of total body heat. The core temperature was evaluated with a probe (1 mm diameter) that was gently inserted, after lubrication with liquid Vaseline, into the rectum of the mouse (to about 1 cm) and left in position until the stabilization of the temperature (about 10 s) [33]. The probe was connected to a Cole Parmer digital thermometer, model 8402. Stress was equalized to a normal routine clinical procedure.

#### 4.5.5. Evaluation of Pain Induced by Mechanical Stimulation of the Tail

The tail pinch test is used for evaluating acute mechanical nociception [33]. A special rigid probe connected to a digital dynamometer (ZP-50N, 441-8077, Toyohashi, Aichi, Japan) was gently placed on the tail (in the distal portion), and progressive pressure was applied. When the rat flicked its tail, the pressure was stopped, and the digital instrument saved the maximum peak of weight supported (g/force). A cut-off (500 g/force) was set to avoid tissue damage. The test was repeated three times, and the final value was calculated with the average of three obtained scores.

#### 4.5.6. Drag Test

The drag test measures the ability of the animal to balance the body posture with the front legs in response to an externally dynamic stimulus [33]. The mouse was lifted by the tail, leaving the front paws on the table, and dragged backward at a constant speed of about 20 cm/s for a fixed distance (100 cm). The number of steps performed by each paw was recorded by two different observers. For each animal, from five to seven measurements were collected [33].

### 4.6. Statistical Analysis

Data are expressed in arbitrary units (visual objects and acoustic responses) and percentage of baseline (visual placing response, breath rate, and drag test). Core temperature values were expressed as the difference between control temperature (before injection) and temperature following drug administration (Δ °C). Antinociception (tail pinch tests) was calculated as percentage of maximal possible effect:EMax% = [(test − control latency)**/**(cut off time − control)] × 100

All data are shown as mean ± SEM of 8 independent experimental replications. Statistical analysis of the effects of JWH-175 at different concentrations over time was performed by two-way ANOVA followed by Bonferroni post hoc test for multiple comparisons. Statistical analysis of the effects of AM-251 on JWH-175 effects at 6 and 15 mg/kg was performed by one-way ANOVA followed by Tukey post hoc test for multiple comparisons. Correlation between effects induced by JWH-175 and JWH-018 and JWH-018 plasma concentrations were determined by performing Pearson’s correlation calculation. ED50 (dose of agonist to obtain 50% of the overall mean effect) values were calculated by non-linear regression analysis of dose–response data performed using Prism software (GraphPad Prism, San Diego, CA, USA). The calculation of JWH-018 ED50 values was based on the results of previous studies [33,44]. Curves have been compared, when possible, performing the F test (curve comparison). All statistical analysis was performed using the program Prism 8.0 software (GraphPad Prism, San Diego, CA, USA).

### 4.7. In Vitro and In Vivo Metabolic Studies

#### 4.7.1. Chemicals

The reagents (formic acid, acetonitrile, ethyl acetate, sodium phosphate, sodium hydrogen phosphate, potassium carbonate, potassium hydrogen carbonate) all of analytic grade, were from Sigma-Aldrich (Milan, Italy). Water was ultra-purified using a Milli-Q system (Millipore, Vimodrone, Milan, Italy). The human liver microsomes (HLMs, from 20 Caucasian male and female donors of different ages), the cDNA-expressed CYPs (CYP1A2, CYP3A4, CYP3A5, CYP2C9, CYP2C19, and CYP2D6), and all the reagents used for the in vitro metabolism (i.e., sodium phosphate buffer, NADPH regenerating system containing NADP+, glucose-6-phosphate, and glucose-6-phosphate dehydrogenase) were supplied by Corning Incorporated (Milan, Italy). The enzyme **β**-glucuronidase from E. Coli was purchased from Roche (Monza, Italy).

#### 4.7.2. Protocol for the In Vitro Formation of the JWH-018 Phase I Metabolites and for the Characterization of the Enzymatic Isoforms Involved

JWH-018 was incubated in the presence of HLMs for the enzymatically assisted synthesis of phase I metabolites. JWH-175 was incubated in the presence of the cDNA-expressed CYPs selected separately to characterize the enzymatic isoforms involved in its phase I metabolism. The incubation conditions (i.e., substrate concentration, enzymatic protein concentration, and incubation time) were optimized from protocols already in use in our laboratory to perform metabolism studies [97,98,99]. All incubations were performed in phosphate buffer (0.1 M, pH 7.4). The final incubation medium contained 40 µM of substrate, 0.5 mg/mL of proteins (HLMs or the cDNA-expressed CYPs), 3.3 mM of magnesium chloride, 1.3 mM of NADP+, 3.3 mM of glucose-6-phosphate, and finally 0.4 U/mL of glucose-6-phosphate dehydrogenase. Samples were incubated at 37 °C for different intervals of time. After incubation, 250 μL of acetonitrile was added to stop the phase I reactions. The samples were then pretreated using the analytical procedures currently adopted by our laboratory to detect JWH-018 and its metabolites in human urine during doping control tests.

#### 4.7.3. Protocol for the In Vivo Pharmacokinetic Studies and Behavioral Correlation

To evaluate the in vivo metabolism of JWH-175 and to correlate the pharmacological effects of JWH-175 and JWH-018 (both at 10 mg/kg; i.p.) with plasmatic levels of JWH-018, we performed behavioral analyses coupled with mouse plasmatic samples. In particular, somatosensory responses (visual object, visual placing, and acoustic), body temperature, breath rate, mechanical analgesia, and motor activity (drag test) were detected in mice at timed intervals after JWH-175 and JWH-018 injection (0, 30, 180 and 300 min) based on behavioral test results, and immediately after these measures, a series of blood specimens was collected for quantitative analysis of JWH-175 and JWH-018 (at each considered time point). Moreover, in a further group of mice, urine samples were collected after vehicle or JWH-175 10 mg/kg injection.

##### Sample Collection

For the in vivo studies, three different groups of mice were selected. For the first group, pooled urine samples were collected in a range of 0–6 h after the injection of JWH-175 (10 mg/kg). Urine specimens were collected from mice individually placed inside metabolic cages (Ugo Basile SRL, Gemonio (VA), Italy) with free access to water and food [97,100,101]. The same dose of JWH-175 was administered to the second group, and behavioral tests were performed before the blood collection. Tests and sampling were carried out, at 0, 30, 60, and 300 min, on a sub-group of six mice at each time interval. The same behavioral studies and blood sampling were performed on the third group, to which JWH-018 (10 mg/kg) was injected. Blood samples (mean total volume: 500 μL) were collected by submandibular blood collection technique into 1 mL vials containing EDTA (4 mg/mL of blood) as preservative and anticoagulant. After each blood withdrawal, an equal volume of saline solution was subcutaneously injected into mice to maintain volume and osmotic homeostasis. Plasma from collected blood and urine samples were stocked at −20 °C until the analysis.

##### Sample Pre-Treatment

For blood metabolic profile, samples were centrifugated at 9000× *g* rpm for 8 min to obtain plasma. Then, 200 µL of the supernatant was added to 500 μL of acetonitrile and centrifugated at 13,000× *g* rpm for 3 min. The aqueous layer was then collected and added to 50 μL of internal standard solution (10 μg/mL) and 500 μL of phosphate buffer (0.8 M, pH 7.4). Samples were then incubated for 1 h at 55 °C. After hydrolysis, samples were extracted with 5 mL of ethyl acetate. The organic solvent was then dried under nitrogen flow at 40 °C, and the dry residue was resolved in 50 μL of mobile phase and an aliquot of 10 µL was injected into the LC-MS systems.

For the urinary excretion studies, a volume of 50 μL of urine samples was added to 50 μL of internal standard solution (ISTD, JWH210 standard solution final concentration 10 μg/mL) and 20 μL of β-glucuronidase. The samples were then buffered with 200 μL of phosphate buffer (0.8 M, pH 7.4) and extracted with 5 mL of ethyl acetate. The organic solvent was then dried under nitrogen flow at 40 °C, and the dry residue was resolved in 50 μL of mobile phase and an aliquot of 10 µL was injected into the LC-MS systems.

#### 4.7.4. Instrumental Conditions

Samples were analyzed using an Agilent 1200 series high-performance liquid chromatography (HPLC) instrument equipped with a SUPELCO C18 column (15 cm × 2.1 mm × 5 µm) coupled with a Sciex 5500QTRAP triple quadrupole mass spectrometer (Sciex, Milan, Italy) with an ESI source operated in positive ionization mode. Analyses were carried out at a constant flow rate of 250 µL/min using ultra-purified water, 0.1% formic acid (A), and acetonitrile 0.1% formic acid (B) as mobile phase. The gradient started at 5% eluent B, was increased to 65% eluent B in 7 min and after 4 min to 95% eluent B in 2 min, held for 4.5 min, decreased to starting conditions of 5% eluent B in 0.31 min, and held for 2 min for re-equilibration

The mass spectrometric parameters were optimized by infusing the standard solution of JWH-175, JWH-018, and JWH-210 at a concentration of 10 μg/mL. Multiple reaction monitoring (MRM) was used as the acquisition mode (see Table 3). The MRM method was optimized starting from the protocol employed in our laboratory [102].

#### 4.7.5. LC-MS/MS Method Validation

The analytical procedure was validated in terms of specificity, limit of detection (LOD), limit of quantification (LOQ), carryover, linearity, accuracy, ion suppression/enhancement, intraday and interday precision, recovery, and robustness.

The specificity was evaluated by analyzing at least 10 negative samples on two different days to verify that the analytes of interest were effectively differentiated from endogenous matrix interferences or from other substance(s) present in the negative samples selected or in the reagents/devices used for sample collection and extraction.

Carryover was determined by analyzing negative samples immediately after samples containing the compounds of interest at a concentration at least 20 times the LOD.

The effect of the matrices under investigation on the ion suppression and ion enhancement was assessed by comparison of the area of the signals obtained in the negative samples spiked with the compounds under investigation with those obtained in the reconstitution solution (mobile phase initial composition) containing the analytes of interest at the same concentration.

The intraday precision and the interday precision (evaluated on three different days) were determined on two batches of five different negative samples spiked with the compounds under investigation at low (corresponding to the LODs), medium (corresponding to 5 times LODs), and high concentrations (corresponding to 10 times LODs) at the beginning of sample preparation. Both intraday precision and interday precision of the relative retention time (RRT) and of the relative response of each analyte were expressed as CV (%).

Calibration and quality control (QC) samples were prepared by adding the appropriate volume of stock solutions of the compounds of interest (prepared in methanol at 10 µg/mL, 1 µg/mL, and 0.1 µg/mL) to the drug-free blood samples.

Calibration samples were prepared at six concentration levels: 5, 10, 50, 100, 200, 500, and 1000 ng/mL; quality control (QC) samples were instead prepared at three concentration levels: 10, 200, and 800 ng/mL.

The linearity was evaluated considering the coefficient of determination (r2). Limits of detection (LODs) and quantitation (LOQs) were defined as the concentrations yielding signal-to-noise ratios higher than 3 and 10, respectively.

The accuracy and precision of the method were determined by the analysis of QC samples at three different concentrations (10, 200, and 800 ng/mL). Intraday precision was expressed as the relative standard deviation (RSD) (%) of the estimated concentrations obtained for six replicates of the QC samples at the three different concentrations analyzed on the same day. Intermediate precision is given as the RSD (%) of the estimated concentrations obtained for three replicates of the QC samples over 5 different days. Accuracy was instead evaluated by the relative error (%) in the estimation of the concentration for the QC samples.

The recovery of all analytes was estimated by preparing: (i) samples (pre-spiked) using negative samples fortified with the target analytes at low (corresponding to the LOD), medium (corresponding to 5 times the LOD), and high concentrations (corresponding to 10 times the LOD) and (ii) samples (post-spiked) using negative samples spiked with the same concentrations as the pre-spike samples after extraction. The extraction recovery (%) was then calculated by comparing the peak area ratio of the compounds and the peak area of the internal standard of the two sets (pre-spike and post-spike) of samples. The internal standard was added after sample pre-treatment in both sets of samples.

The robustness of the method was evaluated by analyzing negative samples spiked with the analytes of interest at the LOD concentration. The samples were prepared and analyzed once a week for 7 weeks, randomly changing the instrument employed in routine analyses and the operator involved in the instrumental analysis and in the preparation of the samples.

## 5. Conclusions

The present study states the importance of carefully investigating the metabolism of SCBs in order to understand both their possible bio-activation to more potent and toxic compounds, and to identify metabolites to be considered as markers of intake in biological sample analyses, thus highlighting the great toxicological and forensic potential of our results. To this end, the use of human liver microsome and receptor models supports the aim of providing data associated with a more targeted translation to human toxicity. In fact, for the first time, the present study evaluated in vitro and in vivo pharmacodynamic and pharmacokinetic activity of JWH-175 and compared it with its analog JWH-018. Noteworthily, it firstly demonstrated that JWH-175 is rapidly metabolized in vivo to the more potent JWH-018 and that this bio-transformation is directly correlated with the behavioral and physiological changes occurring in mice. In particular, JWH-175 retains nanomolar affinity for both CD-1 murine and human CB1 and CB2 receptors and reduced fEPSP in hippocampal slices. Furthermore, this compound altered visual and acoustic sensorimotor responses, core temperature, breath rate, and motor performance and induced mechanical analgesia in mice possibly via acting on CB1 receptors. The increase in efficacy of JWH-175 in in vivo assays compared to in vitro ones strengthens the hypothesis that behavioral and physiological alterations observed after JWH-175 administration were prevalently dependent on JWH-018 formation in the blood. This hypothesis is also supported by analytical data reporting the presence of JWH-018 and only traces of JWH-175 in the brain tissue of mice systemically treated with JWH-175 (data not shown).

Data from SCB-related emergency department visits suggest a higher risk of intoxication for male patients when compared to female patients [103]. Thereby, in line with a previous study on other SCBs [104], only male mice were used. This aspect can represent a weakness, because of the sex-related differences underlined by further preclinical studies on NPSs and other drugs of abuse [105]. Nevertheless, the present results also offer a starting point for future gender-based direction studies.

## Figures and Tables

**Figure 1 ijms-23-08030-f001:**
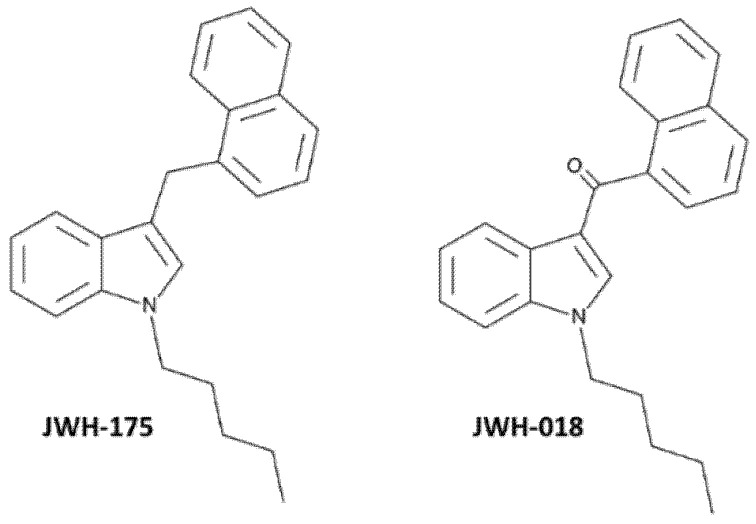
Chemical structures of JWH-175 (3-(1-naphthalenylmethyl)-1-pentyl-1H-indole) and JWH-018 (1-naphthalenyl(1-pentyl-1H-indol-3-yl)-methanone) copied from the Cayman Chemical website (https://www.caymanchem.com, accessed date 17 April 2022).

**Figure 2 ijms-23-08030-f002:**
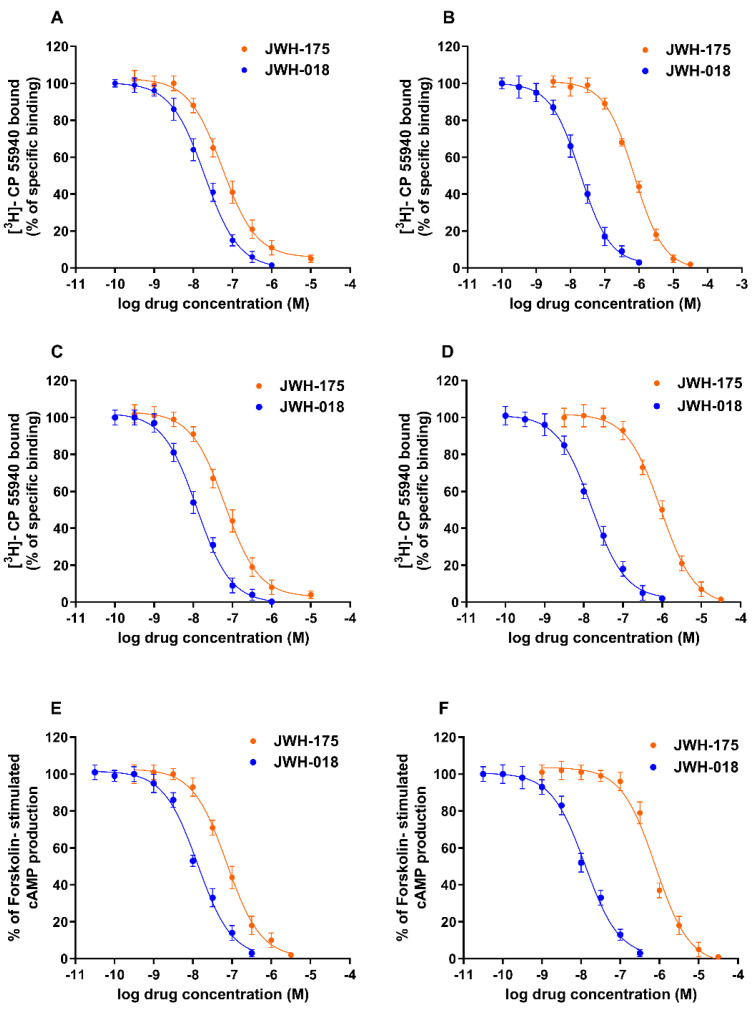
Competition curves of specific [3H]-CP 55,940 binding by synthetic cannabinoids JWH-175 and JWH-018 in CHO cell membranes transfected with human CB1 receptors (**A**) or human CB2 receptors (**B**) and to CB1 receptors expressed in mouse brain membranes (**C**) or CB2 receptors expressed in mouse spleen membranes (**D**). Inhibition curves of forskolin-stimulated cAMP accumulation by JWH-175 and JWH-018 in CHO cells transfected with human CB1 receptors (**E**) or human CB2 receptors (**F**). Results are given as the mean ± SEM of three independent experiments performed in duplicate. Cyclic AMP experiments were performed to evaluate the potency of the examined compounds in CHO cells transfected with human CB1 or CB2 receptors. JWH-175 was 5.38 times less potent at the human CB1 receptors than the reference compound JWH-018 (Table 1). In line with binding experiments, JWH-175 showed a higher potency for CB1 than CB2 receptors. All the tested compounds were able to completely inhibit the forskolin-stimulated cAMP production, thus behaving as full agonists.

**Figure 3 ijms-23-08030-f003:**
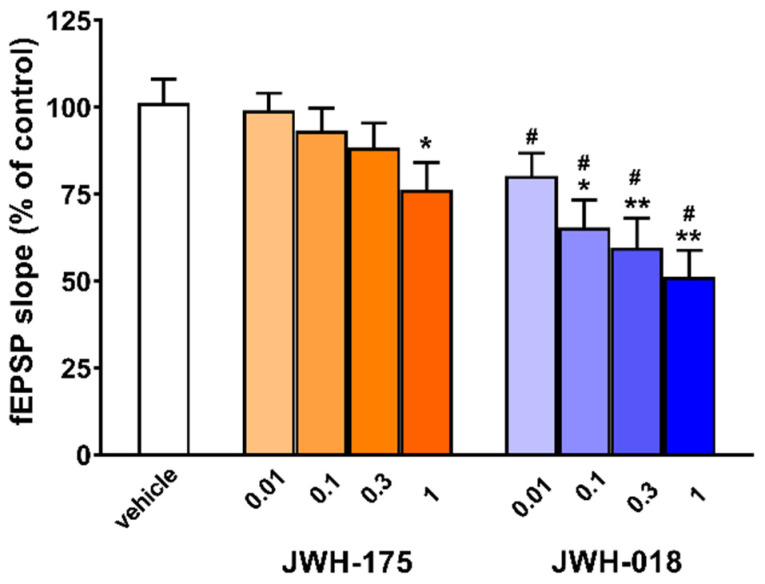
Effect of JWH-175 (0.01–1 µM) and JWH-018 (0.01–1 µM) on fEPSP of CA1 area of mouse hippocampal slice. Histogram reporting average effects at steady state, as average fEPSP slope of last 3 min of recording. Values correspond to % changes compared to control condition (average values of fEPSP slope 10 min before drug application). The analysis of the mean overall effect of each compound was performed with one-way ANOVA followed by Tukey’s test. * *p* < 0.05, ** *p* < 0.01 versus vehicle. Comparisons between JWH-018 and JWH-175 at the same concentrations were performed with unpaired *t* test. # *p* < 0.01 versus JWH-018.

**Figure 4 ijms-23-08030-f004:**
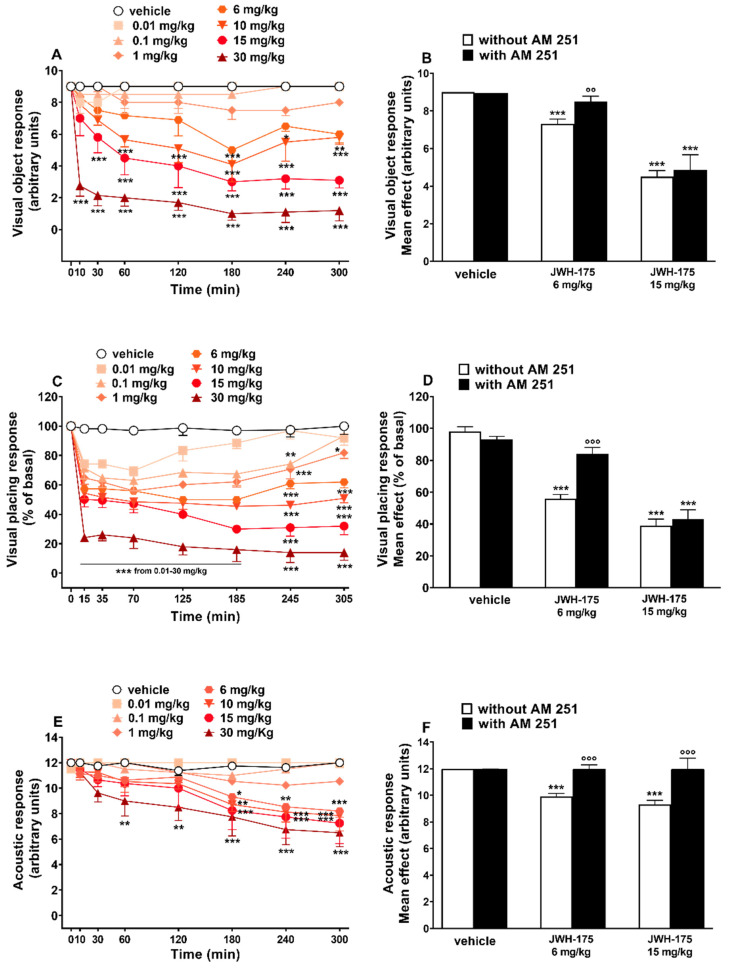
Effect of JWH-175 (0.01–30 mg/kg i.p.) on the visual object test (**A**), visual placing test (**C**), and acoustic response (**E**) in mice. Interactions of JWH-175 with the selective CB1 receptor antagonist AM-251 (6 mg/kg i.p.) are reported in histograms (**B**,**D**,**F**). Data are expressed as arbitrary units (visual object and acoustic responses) or as a percentage of baseline (visual placing response) and represent the mean ± SEM of 8 animals for each treatment. Statistical analysis was performed by two-way ANOVA followed by Bonferroni’s test (**A**,**C**,**E**) for multiple comparison for the dose–response curve of each compound at different time-points. The analysis of the mean overall effect of each compound and AM-251 (**B**,**D**,**F**) was performed with one-way ANOVA followed by Tukey’s test. * *p* < 0.05, ** *p* < 0.01, *** *p* < 0.001 versus vehicle; °° *p* < 0.01, °°° *p* < 0.001 versus JWH-175.

**Figure 5 ijms-23-08030-f005:**
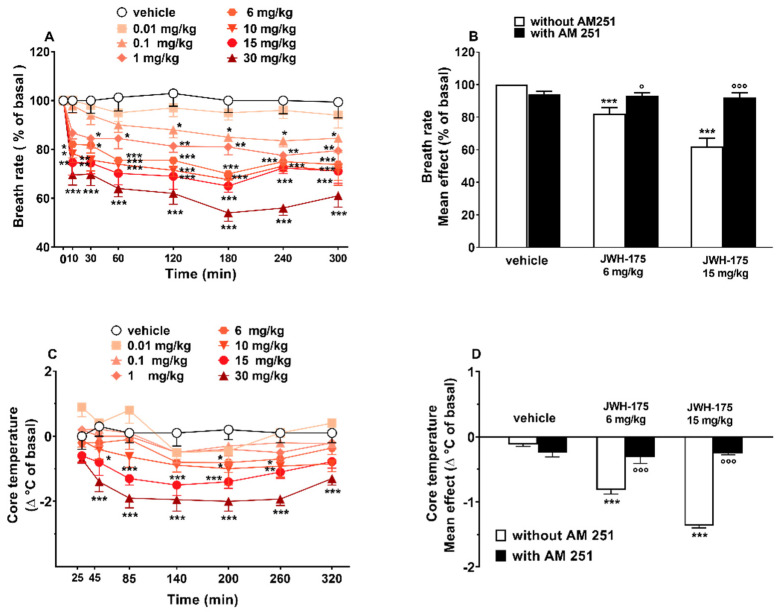
Effect of JWH-175 (0.01–30 mg/kg i.p.) on the breath rate (**A**) and core body temperature (**C**) in mice. Interactions of JWH-175 with the selective CB1 receptor antagonist AM-251 (6 mg/kg i.p.) are reported in histograms (**B**,**D**). Data are expressed as percentage of baseline (breath rate) and as the difference between control temperature (before injection) and temperature following drug administration (Δ°C of basal) and represent the mean ± SEM of 8 animals for each treatment. Statistical analysis was performed by two-way ANOVA followed by Bonferroni’s test (**A**,**C**) for multiple comparison for the dose–response curve of each compound at different time-points. The analysis of the mean overall effect of each compound and AM-251 (**B**,**D**) was performed with one-way ANOVA followed by Tukey’s test. * *p* < 0.05, ** *p* < 0.01, *** *p* < 0.001 versus vehicle; ° *p* < 0.05, °°° *p* < 0.001 versus JWH-175.

**Figure 6 ijms-23-08030-f006:**
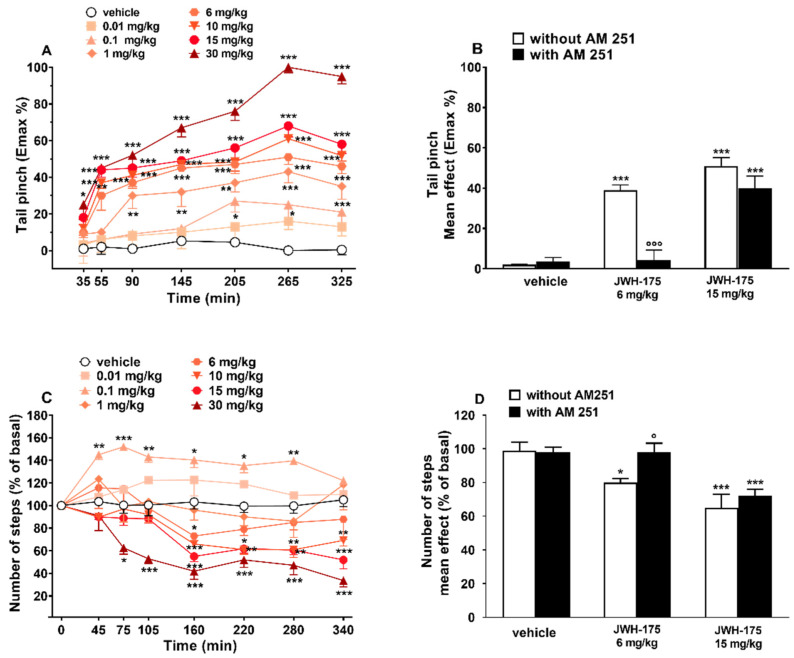
Effect of JWH-175 (0.01–30 mg/kg i.p.) on the tail pinch test (**A**) and drag test (**C**) in mice. Interactions of JWH-175 with the selective CB1 receptor antagonist AM-251 (6 mg/kg i.p.) are reported in histograms (**B**,**D**). Data are expressed as percent of maximum effect (Emax %; tail pinch test) and as percent of baseline (drag test) and represent the mean ± SEM of 8 animals for each treatment. Statistical analysis was performed by two-way ANOVA followed by Bonferroni’s test (**A**,**C**) for multiple comparison for the dose–response curve of each compound at different time-points. The analysis of the mean overall effect of each compound and AM-251 (**B**,**D**) was performed with one-way ANOVA followed by Tukey’s test. * *p* < 0.05, ** *p* < 0.01, *** *p* < 0.001 versus vehicle; ° *p* < 0.05, °°° *p* < 0.001 versus JWH-175.

**Figure 7 ijms-23-08030-f007:**
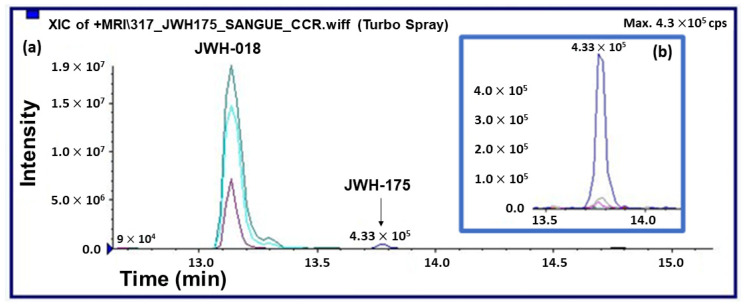
Extracted chromatogram of a representative plasma sample collected from mice after 30 min by the intake of JWH-175 (**a**). Chromatogram enlargement of the plasma level section of JWH-175 (**b**).

**Figure 8 ijms-23-08030-f008:**
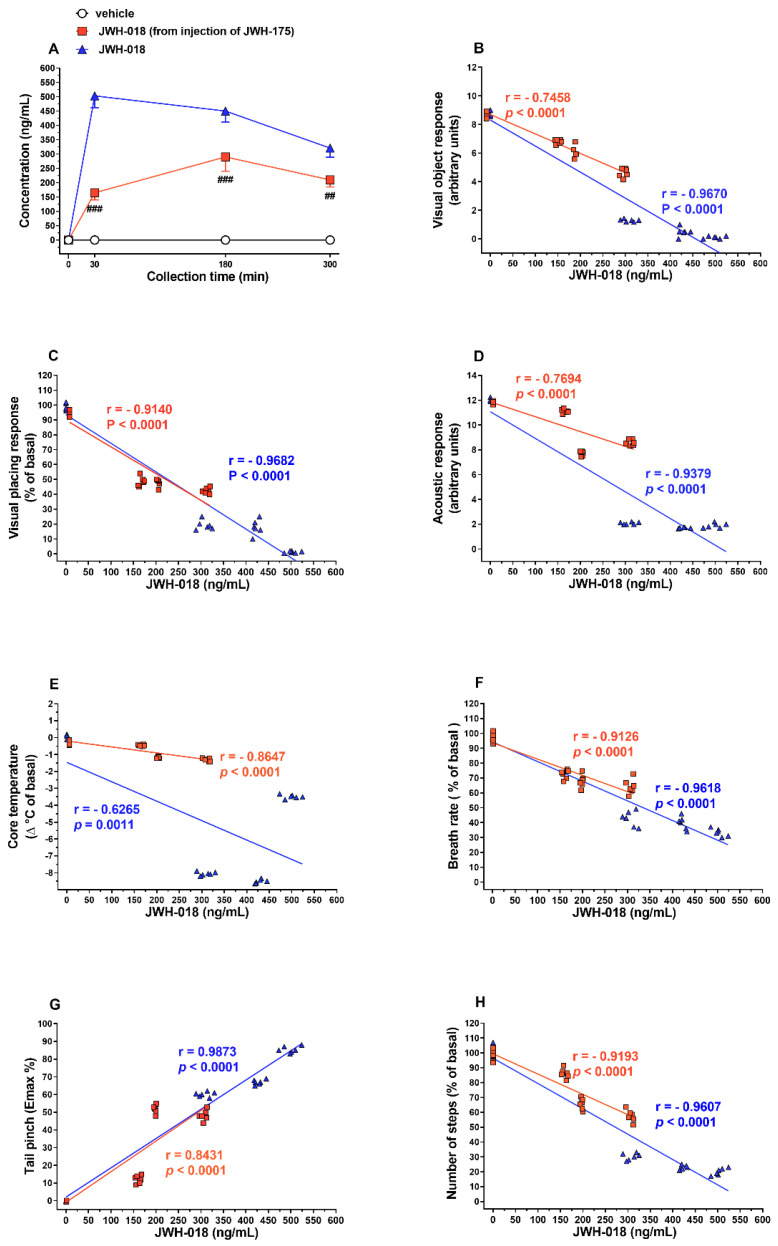
Concentration of JWH-018 in the plasma of mice detected at 30, 180, and 300 min after the injection of JWH-175 (10 mg/kg i.p.) and JWH-018 (10 mg/kg i.p.; (**A**)). Results are expressed as ng/mL and represent the mean ± SEM of 6 samples for each time point. Correlation between visual object (**B**), visual placing (**C**) and acoustic (**D**) responses, body temperature (**E**), breath rate (**F**), mechanical analgesia (**G**), and motor activity (**H**) changes with JWH-018 plasma concentrations. Correlation coefficients were determined by performing Pearson’s correlation calculation. ## *p* < 0.01, ### *p* < 0.001 versus JWH-018.

**Figure 9 ijms-23-08030-f009:**
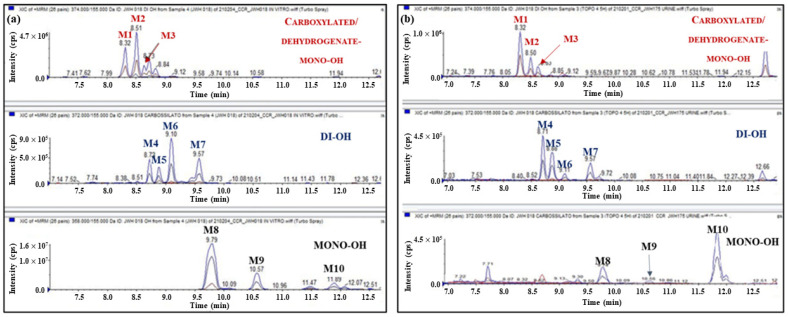
Extracted chromatograms obtained from the analysis of a representative urine sample collected from mice after JWH-175 administration (**a**) and from the analysis of samples obtained by incubating JWH-018 in the presence of HLMs (**b**). The phase I metabolites in common between JWH-018 and JWH-175 are three di-hydroxylated metabolites (M1–M3), four dehydrogenate-mono-OH metabolites (M4–M7), and three mono-hydroxylated metabolites (M8–M10).

**Table 1 ijms-23-08030-t001:** Binding and functional parameters of JWH-175 at human and mouse CB1 and CB2 receptors, in comparison to JWH-018.

Compound	hCB_1_ CHO Membranes ^a^ Ki (nM)	hCB_2_ CHO Membranes ^a^ Ki (nM)	Mouse Cortex Membranes CB_1_ ^a^ Ki (nM)	Mouse Spleen Membranes CB_2_ ^a^ Ki (nM)	hCB1 CHO Cells ^b^ IC_50_ (nM)	hCB2 CHO Cells ^b^ IC_50_ (nM)
**JWH-175**	25.8 ± 1.9	363 ± 31	33.6 ± 2.4	487 ± 39	72 ± 5	864 ± 61
**JWH-018 ***	9.52 ± 0.73	8.63 ± 0.69	5.82 ± 0.44	7.13 ± 0.53	13.69 ± 1.04	11.62 ± 0.97

Data are expressed as mean ± SEM. ^a^ [3H]-CP-55,940 competition binding experiments. ^b^ Cyclic AMP experiments. * Data elaborated from [15,33].

**Table 2 ijms-23-08030-t002:** ED50 values of JWH-175 in comparison to JWH-018. Data represent ED50 + SEM. ED50 has been calculated by non-linear regression curve fitting of the dose–response curves determined using Prism 8.0 software (GraphPad Prism, San Diego, CA, USA). Curves have been compared, when possible, by performing F test (curve comparison). * *p* < 0.05, ** *p* < 0.01, *** *p* < 0.001 versus JWH-018. ^#^ JWH-018 data were elaborated from [33,34].

Test	JWH-175 ED_50_ (mg/kg)	JWH-018 ^#^ ED_50_ (mg/kg)
**Visual Object**	9.6 ± 0.02 ***	0.32 ± 0.06
**Visual Placing**	5.2 ± 0.08 ***	0.7 ± 0.06
**Startle Reflex**	4.0 ± 0.03 ***	0.53 ± 0.05
**Breath Rate**	1.7 ± 0.15	N/D
**Core Temperature**	5.6 ± 0.04 ***	1.7 ± 0.04
**Tail Pinch**	4.0 ± 0.12 *	1.8 ± 0.11
**Drag Test**	10.9 ± 0.12 **	2.4 ± 0.12

**Table 3 ijms-23-08030-t003:** Mass spectra parameters for metabolism studies including precursor and product ions and relative collision energies for QqQ.

Compound	Precursor Ions (*m*/*z*)	Product Ions (*m/z*)	Collision Energy (eV)
JWH-210 (ISTD)	**370**	183	30
JWH-175	**370**	155; 144; 127; 200	35; 40; 40; 30
JWH-018	**342**	155; 144; 127; 200; 216	35; 40; 40; 30; 30
- Mono-OH	**358**	155; 127; 144; 200; 237	35; 40; 40; 30; 30
- Di-OH	**374**	155; 144; 127; 200; 230	35; 40; 40; 30; 30
- Carboxylate - Dehydrogenate-mono-OH	**372**	155; 144; 127; 200; 230	35; 40; 40; 30; 30

## Data Availability

The data presented in this study are available on request from the first (Micaela Tirri) and corresponding author (Matteo Marti) for researchers of academic institutes who meet the criteria for access to the confidential data.

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
