# Peer review of "In Vivo Bio-Activation of JWH-175 to JWH-018: Pharmacodynamic and Pharmacokinetic Studies in Mice"

_ijms, 2022, doi:10.3390/ijms23148030_

Round 1

Reviewer 1 Report

Summary:

The current work characterizes the pharmacokinetics (PK) and pharmacodynamics of (PD) JWH-175 (synthetic cannabinoid). The primary objective of this study is to evaluate whether JWH-175 could be bio-transformed in vivo to JWH-018 and to compare the PK/PD profiles of JWH-175 with its more potent counterpart, JWH-018, in mice. The results showed that JWH-175 is quickly bio-activated into JWH-018 in the blood. This suggests that the in vivo PD effects of JWH-175 are likely due to the formation of JWH-018, and in vivo PD experiments showed that JWH-175 is less potent than JWH-018. The goal of the manuscript is clearly defined, the experimental design is appropriate, the title adequately describes the content of the publication, and the results support the conclusion. This work is relevant to the scope of the journal. However, there is a opportunity to further improve the manuscript by including the points listed below.

Major comments:

1.     Introduction section can be strengthened by adding information on dose, and route of administration of select synthetic cannabinoid such as JWH-175 and JWH-018.

2.     Method section shall be added with justification of the following: acute vs chronic administration of JWH-175, mice vs other non-clinical model, and PK/PD sampling time point (duration).

3.     There is a need to add text regarding study strength and limitations (e.g., only male mice were tested).

4.     Please provide supporting data if the current findings can be extrapolated from mice to human along with recommendations on future directions.

Minor comments:

11.  It is suggested to provide high-resolution figures.

22.  Line 100: Is it human microsomes or hepatocytes model?

33. The author should include full names to define the abbreviated terms throughout the text.

44.  Line 156 (… were reported in histograms (B, D, E)..):  Is it “F” or “E”?

55.  Line 165: The parenthesis statement "F6,392 = 147.8" is unclear, and the same wording has been noticed elsewhere. Please provide further clarification.

Author Response

Response to Reviewer 1

We thank the Reviewer 1 for his/her evaluation of our manuscript and for helpful concerns to improve the article. In this revised version of the work we have addressed the major concerns of the referee (highlighted in yellow).

Major comments

Rev1Q1: Introduction section can be strengthened by adding information on dose, and route of administration of select synthetic cannabinoid such as JWH-175 and JWH-018.

AA: We thank the Reviewer 1 for this suggestion and have added information about dose and route of administration of the tested compounds in the introduction section. 

Rev1Q2: Method section shall be added with justification of the following: acute vs chronic administration of JWH-175, mice vs other non-clinical model, and PK/PD sampling time point (duration).

AA: We thank the Reviewer 1 for this suggestion and have provide this information in materials and methods section.

Rev1Q3: There is a need to add text regarding study strength and limitations (e.g., only male mice were tested).

AA: We thank the Reviewer 1 for this suggestion and have added strength and limitations in the conclusions section. 

Rev1Q4: Please provide supporting data if the current findings can be extrapolated from mice to human along with recommendations on future directions.

AA: We thank the Reviewer 1 for this suggestion and have specified also this point in the conclusions section. 

Minor comments

Rev1Q11: It is suggested to provide high-resolution figures.

AA: Excusing about this inaccuracy, we have replaced figures with high-resolution versions.

Rev1Q22: Line 100: Is it human microsomes or hepatocytes model?

AA: Excusing about this inaccuracy, we have specified in the sentence that it is human liver microsomes model.

Rev1Q33: The author should include full names to define the abbreviated terms throughout the text.

AA: Excusing about this inaccuracy, we have included full names to define the abbreviated terms in each specific text section.

Rev1Q44: Line 156 (… were reported in histograms (B, D, E)..):  Is it “F” or “E”?

AA: Excusing about this inaccuracy, we have replaced the letter E with the F in the specific text section.

Rev1Q55: Line 165: The parenthesis statement "F6,392 = 147.8" is unclear, and the same wording has been noticed elsewhere. Please provide further clarification.

AA: Excusing about these inaccuracies, we have standardized the parenthesis statements along the text in the results section. 

Reviewer 2 Report

The manuscript presents the results of in vitro and in vivo studies on a synthetic cannabinoid, JWH-175 and its metabolite JWH-018. The study provides some new, interesting data, however, several issues need clarification:

It is not clear why human, not murine microsomes were used to compare metabolite profiles in vitro and in vivo (Fig. 9). The in vivo study was performed on mice. Similarly, it has not been explained, why JWH-018 and not JWH-175 was used in in vitro metabolic studies (section 4.7.2).

Why the dose of 10 mg/kg was selected for pharmacokinetic studies, whereas such dose was not used in behavioral tests.

It is not sure that the maximum plasma concentration occurs at 30 min (Fig. 8 A). The samples should be collected in the absorption phase to properly evaluate Cmax. Moreover, pharmacokinetic parameters of both studied compounds should be estimated. The samples should be collected for a longer period of time to calculate elimination rate constants.

The authors used a linear model for all effect-concentration relationships (Figure 8). When looking on the effect-time relationships (Figures 4-6), the maximum responses are delayed in relation to the maximum plasma concentrations in most cases. Therefore, using direct response models, such as the linear model, is not appropriate.

It has not been explained whether the unchanged parent compound and its main metabolite were present in mouse urine.

The enzymes involved in metabolism of both studied compounds should be identified.

The number of animals/samples pre group should be indicated in figure captions.

The authors state (p. 15, line 449) that JHW-175 after 30 min was not detectable, whereas the area showed in Fig. 2 – 4.33e5 is quite large. Therefore, the profile of JHW -175 should be presented as well.

The route of compound administration and the volume of blood collected was not indicated (section 4.7.3.1)

The volume of plasma used for analysis was not indicated (section 4.7.3.2).

The ratio at which the mobile phase components were mixed was not shown (section 4.7.4.).

The method validation parameters (i.e. precision, accuracy, LOQ, linearity range) should be shown.

In the results section the author used plasma and blood concentrations exchangeably.  It does not mean the same. Blood and plasma concentrations may differ.

Author Response

Response to Reviewer 2

We thank the Reviewer 2 for his/her evaluation of our manuscript and for helpful concerns to improve the article. In this revised version of the work we have addressed the major concerns of the referee (highlighted in blue).

Major comments

Rev2Q1: It is not clear why human, not murine microsomes were used to compare metabolite profiles in vitro and in vivo (Fig. 9). The in vivo study was performed on mice. Similarly, it has not been explained, why JWH-018 and not JWH-175 was used in in vitro metabolic studies (section 4.7.2).

AA: In vitro studies were performed to synthetize enzymatically the main metabolites of JWH018, being the reference materials of these metabolites not available in our laboratory. We did not perform the in vitro studies to characterize the metabolic profile of JWH175, because these studies were already performed by previous authors.

We have used human liver microsomes instead of murine microsomes to demonstrate that the metabolic profile in human and in mice is qualitatively equivalent.

Rev2Q2: Why the dose of 10 mg/kg was selected for pharmacokinetic studies, whereas such dose was not used in behavioral tests.

AA: We thank the Reviewer 2 for pointing out this aspect of the study. In particular, the present results aim to characterize pharmacokinetic and pharmacodynamic profile of JWH-175 and compare them with those of JWH-018. Based on our previous studies on JWH-018 (Vigolo et al., 2015; Ossato et al., 2015), this compound can induce severe neurological adverse effects (i.e. myoclonias or seizures) at lower dosages respect to JWH-175. Thereby, we have considered 10 mg/kg as the maximal dosage to use in order to follow ARRIVE guidelines for animals’ care and detect a considerable plasma concentration of JWH-018 and JWH-175 from injection of JWH-175. Furthermore, the somato-sensory responses (visual object, visual placing and acoustic), body temperature, breath rate, mechanical analgesia and motor activity (drag test) induced by the systemic administration of 10 mg/kg of JWH-175 were detected in mice for both behavioural and pharmacokinetic studies (Figure 8). However, many of these results have been presented in Figure 8 relative to selected specific time points as responses related to the plasma concentration. Therefore, we avoided to report the same results in dose-response curves in order to simplify the graphical presentation and the repetition of the same data. However, we agree with Reviewer 2 and have added data relative to dosage of 10 mg/kg in behavioural responses’ graphs to clarify this point.

Rev2Q3: It is not sure that the maximum plasma concentration occurs at 30 min (Fig. 8A). The samples should be collected in the absorption phase to properly evaluate Cmax. Moreover, pharmacokinetic parameters of both studied compounds should be estimated. The samples should be collected for a longer period of time to calculate elimination rate constants.

AA: We agree, we have changed kinetic with profile of the compounds of interest in blood.

Rev2Q4: The authors used a linear model for all effect-concentration relationships (Figure 8). When looking on the effect-time relationships (Figures 4-6), the maximum responses are delayed in relation to the maximum plasma concentrations in most cases. Therefore, using direct response models, such as the linear model, is not appropriate.

AA: We thank the Reviewer 2 for pointing out this aspect of the study. This aspect is also very attention-grabbing for us. The present study demonstrated the in vivo bioactivation of JWH-175 to its more potent counterpart JWH-018. Specifically, effect-time relationships’ data report the effects induced by JWH-175 injection in mice (Fig. 4-6). On the other hand, plasma concentration presented in effect-concentrations relationships’ graphs refers to concentrations peaks of JWH-018 derived from the injection of JWH-175 at each specific time point. Therefore, we consider it as the possible explanation of the delayed responses observed in dose-response curves. Due to the fact that the increase in plasma concentrations of JWH-018 coincides with a worsening of the effects, we believe that the direct correlation model can be adequate to describe the present results (in line with previous studies carried out on SCBs in our laboratory; Bilel et al., 2019).

Bilel, S., Tirri, M., Arfè, R., Stopponi, S., Soverchia, L., Ciccocioppo, R., Frisoni, P., Strano-Rossi, S., Miliano, C., De-Giorgio, F., Serpelloni, G., Fantinati, A., De Luca, M. A., Neri, M., & Marti, M. (2019). Pharmacological and Behavioral Effects of the Synthetic Cannabinoid AKB48 in Rats. Frontiers in neuroscience, 13, 1163. https://doi.org/10.3389/fnins.2019.01163

Rev2Q5: It has not been explained whether the unchanged parent compound and its main metabolite were present in mouse urine.

AA: We showed both parent compound and metabolites to confirm the formation of JWH-018 from JWH-175.

Rev2Q6: The enzymes involved in metabolism of both studied compounds should be identified.

AA: Our in vitro investigation indicated that the isoenzymes involved in the formation of JWH/018 from JWH/175 are CYP3A4 and CYP3A5. The isoenzymes involved in the formation of the hydroxylated metabolites are CYP3A4, CYP3A5 and in minor amount also CYP2D6, CYP2C19 and CYP2C9. JWH-175 is the parent compound.

Rev2Q7: The number of animals/samples pre group should be indicated in figure captions.

AA: Excusing about this inaccuracies, we provide information about the number of animals/samples in figure captions.

Rev2Q8: The authors state (p. 15, line 449) that JHW-175 after 30 min was not detectable, whereas the area showed in Fig. 2 – 4.33e5 is quite large. Therefore, the profile of JHW -175 should be presented as well.

AA: At 180 min JWH-175 was not detectable. The signal with area 4.33e5 corresponds to few ng/mL. Consequently, the profile of JWH175 in blood cannot be reported.

Rev2Q9: The route of compound administration and the volume of blood collected was not indicated (section 4.7.3.1).

AA: Excusing about this inaccuracies, we provide information about route of administration and volume of collected samples in the specific section of the manuscript.

Rev2Q10: The volume of plasma used for analysis was not indicated (section 4.7.3.2).

AA: This information was added.

Rev2Q11: The ratio at which the mobile phase components were mixed was not shown (section 4.7.4.).

AA: The elution gradient was added.  

Rev2Q12: The method validation parameters (i.e. precision, accuracy, LOQ, linearity range) should be shown.

AA: The conditions and results of method validation were added.

Rev2Q13: In the results section the author used plasma and blood concentrations exchangeably.  It does not mean the same. Blood and plasma concentrations may differ.

AA: Excusing about this inaccuracies, we have corrected and specified the samples type.

Reviewer 3 Report

The word “respect” has been used incorrectly for couples of times in the manuscript. The meaning of “respect” is totally different to “in respect to”. The sentences should be rephrased correctly.

The authors compared activities of the synthetic cannabinoids JWH-175, the prodrug and its active metabolite JWH-018 in vitro and their behavioral activity in vivo. JWH-175 shows more specificity to CB1 than to CB2 receptors (Ki: 15.8 vs 363 nM), while JWH-018 shows no specificity to neither CB1 nor CB2 (Ki: 9.52 vs 8.63 nM). Nevertheless, JWH-018 is more potent than JWH-175. In vivo, JWH-175 is extensively metabolized to JWH-018. In plasma, the levels of JWH-175 is almost non-existent, and the levels of JWH-175 in brain tissues have not been measured. It is anticipated that the levels of JWH-175 could be too low to exert any effects. In the vivo study, even though a CB1 receptor antagonist AM-251 was used to differentiate the respective CB1 and CB2 effects, the experiment still only evaluates the JWH-018 effects. The observed effects in vivo is likely due to the different levels of JWH-018 after administration of the respective JWH-175 and JWH-018.

The conclusion can only be valid if the authors could confirm the presence of JWH-175 in the respective brain tissue.

Author Response

Response to Reviewer 3

We thank the Reviewer 3 for his/her evaluation of our manuscript and for helpful concerns to improve the article. In this revised version of the work we have addressed the major concerns of the referee (highlighted in green).

Rev3Q1: The word “respect” has been used incorrectly for couples of times in the manuscript. The meaning of “respect” is totally different to “in respect to”. The sentences should be rephrased correctly.

AA: Excusing about this inaccuracies, we have corrected the word “respect” to “in respect to” or other sentences that can correctly explain the meaning.

Rev3Q2: The authors compared activities of the synthetic cannabinoids JWH-175, the prodrug and its active metabolite JWH-018 in vitro and their behavioral activity in vivo. JWH-175 shows more specificity to CB1 than to CB2 receptors (Ki: 15.8 vs 363 nM), while JWH-018 shows no specificity to neither CB1 nor CB2 (Ki: 9.52 vs 8.63 nM). Nevertheless, JWH-018 is more potent than JWH-175. In vivo, JWH-175 is extensively metabolized to JWH-018. In plasma, the levels of JWH-175 is almost non-existent, and the levels of JWH-175 in brain tissues have not been measured. It is anticipated that the levels of JWH-175 could be too low to exert any effects. In the vivo study, even though a CB1 receptor antagonist AM-251 was used to differentiate the respective CB1 and CB2 effects, the experiment still only evaluates the JWH-018 effects. The observed effects in vivo is likely due to the different levels of JWH-018 after administration of the respective JWH-175 and JWH-018.

The conclusion can only be valid if the authors could confirm the presence of JWH-175 in the respective brain tissue.

AA: We thank the Reviewer 3 for pointing out this aspect of the study which is also very attention-grabbing for us. The present study demonstrated only the in vivo bioactivation of JWH-175 to its more potent counterpart JWH-018 in the plasma. However, we confirm the presence of JWH-175 on mice brain tissue. We planned to use this results in a further manuscript.

Round 2

Reviewer 2 Report

The Authors satisfactorily addressed most of my comments and suggestions. It is still not clear whether blood or plasma was analyzed in the pharmacokinetic study. In the Methods section they say:

"Blood and urine samples were stocked at -20 °C until the analysis".

In the Results section "plasmatic concentrations" are presented.

Author Response

Response to Reviewer 2- Round 2

We thank the Reviewer 2 for his/her evaluation of our manuscript and for helpful concerns to improve the article. In this revised version of the work we have addressed the major concerns of the referee (highlighted in pink).

Major comments

Rev2Q1: The Authors satisfactorily addressed most of my comments and suggestions. It is still not clear whether blood or plasma was analyzed in the pharmacokinetic study. In the Methods section they say: "Blood and urine samples were stocked at -20 °C until the analysis". In the Results section "plasmatic concentrations" are presented.

AA: Excusing about inaccuracies, we have specified the sample type also in methods section.

Reviewer 3 Report

The authors have not attempted to answer to the most critical question that JWH-175 is mostly metabolized to JWH-018. The presence of JWH-175 in plasma is almost zero. The amount of JWH-175 in brain is expected to be minimum if present. Therefore, the observed neuro-behavioral effects after administration of JWH-175 are mostly due to JWH-018. The authors suggested that they confirmed the presence of JWH-175 in brain tissue. Then what is the amount of JWH-175 in brain tissue? How is that amount compared to the amount of JWH-175 that can elicit effects in vitro. Without that information, there is no evidence that JWH-175 can cause neuro-behavioral effects in in in vivo especially in the light that JWH-175 is less potent than JWH-018 in in vitro.

There have been couples of studies on the neuro-behavioral effects of JWH-018 in vivo.

Synthetic cannabinoid JWH-018 and its halogenated derivative JWH-018-Cl and JWH-018-Br impair novel object recognition in mice: Behavioral, electrophysiological and neurochemical evidence. Neuropharmacology, 109, 254-269, 2016.

Novel halogenated derivatives of JWH-018: Behavioral and binding studies in mice. Neuropharmacology, 95, 68-82, 2015

Author Response

Response to Reviewer 3

We thank the Reviewer 3 for his/her evaluation of our manuscript and for helpful concerns to improve the article. In this revised version of the work we have addressed the major concerns of the referee (highlighted in red).

Rev3Q1: The authors have not attempted to answer to the most critical question that JWH-175 is mostly metabolized to JWH-018. The presence of JWH-175 in plasma is almost zero. The amount of JWH-175 in brain is expected to be minimum if present. Therefore, the observed neuro-behavioral effects after administration of JWH-175 are mostly due to JWH-018. The authors suggested that they confirmed the presence of JWH-175 in brain tissue. Then what is the amount of JWH-175 in brain tissue? How is that amount compared to the amount of JWH-175 that can elicit effects in vitro. Without that information, there is no evidence that JWH-175 can cause neuro-behavioral effects in in in vivo especially in the light that JWH-175 is less potent than JWH-018 in in vitro.

There have been couples of studies on the neuro-behavioral effects of JWH-018 in vivo.

Synthetic cannabinoid JWH-018 and its halogenated derivative JWH-018-Cl and JWH-018-Br impair novel object recognition in mice: Behavioral, electrophysiological and neurochemical evidence. Neuropharmacology, 109, 254-269, 2016.

Novel halogenated derivatives of JWH-018: Behavioral and binding studies in mice. Neuropharmacology, 95, 68-82, 2015

AA: We apologized with the Reviewer about our incomplete response. We agree with Reviewer about the observed neuro-behavioral effects after JWH-175 administration. In fact, in line with results of plasma analysis, investigation has revealed the presence of JWH-018 and only traces of JWH-175 in brain tissue. Thereby, the in vivo effect may be mostly due to JWH-018. Moreover, our in vitro electrophysiology studies have shown that JWH-175 inhibits fEPSP ten time less potently than JWH-018. Therefore, the traces of JWH-175 in brain tissue may contribute in “minimal” way to the behavioral effects observed.

We already demonstrated the severe neuro-behavioral effects induced by JWH-018 (Barbieri et al, 2016; Vigolo et al., 2015) and therefore in this manuscript, we offer a comparative study about JWH-175 and JWH-018. In the present paper, it has been also demonstrated that JWH-018 is the main metabolite of JWH-175, in an effort to highlight the potential risk related to JWH-175 consumption. As a result, users unaware of this, could incur into very severe effects because of the presence of JWH-018 bioactivated by metabolization of JWH-175.

We have also clarified this point in conclusions section.

Round 3

Reviewer 3 Report

The authors have rewritten the conclusion that reflected the actual findings in their experiments. The paper can be accepted for publication.

There is a minor typo error in Line 302 - "JWH-018 plasmatic levels remain..." - Generally, it is common to state as "JWH-018 plasma levels remain...."